# GOOD ALLOCATIONS FROM BAD ESTIMATES

**Sílvia Casacuberta**[1][*]   **Moritz Hardt**[2,3]
[1]Stanford University
[2]Max Planck Institute for Intelligent Systems, Tübingen
[3]Tübingen AI Center

## ABSTRACT

Conditional average treatment effect (CATE) estimation is the de facto gold standard for targeting a treatment to a heterogeneous population. The method estimates treatment effects up to an error $\epsilon > 0$ in each of $M$ different strata of the population, targeting individuals in decreasing order of estimated treatment effect until the budget runs out. In general, this method requires $O(M/\epsilon^2)$ samples. This is best possible if the goal is to estimate all treatment effects up to an $\epsilon$ error. In this work, we show how to achieve the same total treatment effect as CATE with only $O(M/\epsilon)$ samples for natural distributions of treatment effects. The key insight is that coarse estimates suffice for near-optimal treatment allocations. In addition, we show that budget flexibility can further reduce the sample complexity of allocation. Finally, we evaluate our algorithm on various real-world RCT datasets. In all cases, it finds nearly optimal treatment allocations with surprisingly few samples. Our work highlights the fundamental distinction between treatment effect estimation and treatment allocation: the latter requires far fewer samples.

## 1 INTRODUCTION

Different groups in a population—be it schools, counties, or age brackets—often respond differently to a treatment (Gail & Simon, 1985; Heckman & Robb, 1985; Imbens & Angrist, 1994; Banerjee & Duflo, 2009). This empirical fact has motivated a significant body of work on estimating conditional average treatment effects (CATE), see, e.g., Imbens & Rubin (2015); Athey & Imbens (2016); Athey et al. (2019); Künzel et al. (2019).

A key application of CATE estimation is in welfare-maximizing *treament allocation*: assigning a limited number of treatments to those groups who benefit most from treatment. The optimal treatment allocation selects groups in descending order of treatment effect until the budget runs out. In practice, however, treatment effects first have to be estimated from data, such as the responses from a randomized controlled trial (RCT) on the population. The standard method estimates the treatment effect in each group and allocates in descending order of estimated treatment effects. Good CATE estimates therefore seem to be the necessary first step of treatment allocation. Indeed, estimating a single average treatment effect in one group up to error $\epsilon > 0$ boils down to mean estimation and requires $\Theta(1/\epsilon^2)$ samples. This sample size requirement holds robustly for almost all problem instances with few exceptions. By extension, given $M$ non-overlapping groups, CATE estimation requires $\Theta(M/\epsilon^2)$ samples—as does finding a $(1-\epsilon)$-optimal treatment allocation. And so it appears that the two problems are essentially equivalent, at least in the *worst-case*.

In contrast, we show that for *typical* instances, we can find a $(1-\epsilon)$-optimal allocation with only $O(M/\epsilon)$ samples. Hence, for $M = O(1)$, treatment allocation typically requires quadratically fewer samples than treatment effect estimation. Whereas estimation has a robust quadratic lower bound, we show that the quadratic lower bound for allocation is *brittle*: It's easy to circumvent in theory with natural assumptions and it doesn't arise in any of the real-world datasets we examine. Our results follow from a simple but powerful win-win situation: If the treatment effect in a group is well above or well below the optimal cut-off, we can figure that out with very few samples. Groups close to the threshold, on the other hand, have similar treatment effects. Therefore, we don't lose much if we mix them up. The only bad case arises when all units cluster around the threshold,

---

[*]Work primarily done while interning at the Max Planck Institute for Intelligent Systems.

not too close and not too far. We turn this intuitive observation into a precise instance-dependent upper bound on the sample complexity of near-optimal treatment allocation. The formal argument is delicate, since we don't know the optimal threshold and we can only work from coarse estimates.

In a nutshell, our theory predicts that near-optimal allocation almost always has a linear—not quadratic—dependence on $1/\epsilon$. We thoroughly verify this prediction in five real-world RCT datasets. In all cases, we can find near-optimal allocations with even fewer samples than our upper bound suggests. Fundamentally, our work highlights the stark difference between allocation and estimation: Nearly optimal allocations do *not* necessarily require highly accurate estimates.

## 1.1 OUR CONTRIBUTIONS

Consider a partition of the population into $M$ groups and a budget $K \in \{1, \ldots, M\}$ that allows treating $K$ out of $M$ groups, such as schools, hospitals, or different age brackets. We refer to a group as a *unit* to indicate that for the purpose of treatment allocation we don't distinguish between individuals within a unit—we either treat all or none of the individuals within a unit. We assume that the cost of treating each unit is the same. Our goal is to identify the $K$ units that would most benefit from receiving a specific treatment. Let $\tau(u) \in [0, 1]$ denote the average effect of treatment in unit $u \in \{1, \ldots, M\}$. A treatment allocation $\mathcal{U} \subseteq [M]$ is any subset of $K = |\mathcal{U}|$ units. The *value* achieved by an allocation $\mathcal{U}$ is the total sum $\sum_{u \in \mathcal{U}} \tau(u)$ of treatment effects. An optimal allocation selects the $K$ units with the highest $\tau(u)$ values, breaking ties arbitrarily, achieving value $V^*$.

In this paper, we study the sample complexity of finding a near-optimal allocation. We say that an allocation is $(1 - \epsilon)$-optimal if it achieves a value $V$ that satisfies $V/V^* \geq 1 - \epsilon$. For bounded treatment effects, standard arguments show that we can always find a $(1-\epsilon)$-optimal allocation from $O(M/\epsilon^2)$ samples. Moreover, there is a problem instance that requires $\Omega(M/\epsilon^2)$ many samples. While these well-known bounds settle the worst-case sample complexity, our work shows that the typical sample complexity of treatment allocation is far lower. What matters for our theoretical results is the shape of the distribution of treatment effects. For all reasonably smooth distributions—those that don't put excessive mass on small intervals—we prove that $O(M/\epsilon)$ many samples suffice to get a $(1 - \epsilon)$-optimal allocation.

**Theorem 1** (Informal). *If the distribution of treatment effect values $\tau(u)$ is "smooth", we can obtain a $(1 - \epsilon)$-optimal allocation for any budget $K \leq M$ with $O(M/\epsilon)$ many samples.*

The key insight behind the theorem is that we only need to estimate the treatment effect values $\tau(u)$ up to accuracy $\rho = \Theta(\sqrt{\epsilon})$, rather than $\epsilon$, in order to obtain a near-optimal allocation. Intuitively, the problem of finding an optimal allocation boils down to the problem of deciding, for each unit $u$, whether $\tau(u)$ is above or below the cut-off $\tau_K$, where $\tau_K$ is the smallest treatment effect of any unit in an optimal allocation. For the units that have $\tau(u)$ value far from $\tau_K$, we can determine so by estimating $\tau(u)$ to low accuracy. For the units that are close to the threshold $\tau_K$, mistakenly selecting one unit for another does not change the value of the allocation much.

We show that an accuracy level of $\rho = \Theta(\sqrt{\epsilon})$ strikes the best balance between having the lowest possible level of accuracy while not losing too much value in the allocation. For this balance to hold, we need the number of units around the threshold $\tau_K$ to be reasonable.

We formalize this required notion of "smoothness" through our definition of $\rho$-*regularity*, which requires that any interval of length at least $\rho$ contain at most $O(\rho M)$ units, that is, a multiple of what we would expect under the uniform distribution. Thus, $\rho$-regularity is a weak measure of closeness to the uniform distribution. Many natural distribution families, such as Gaussians or Beta distributions, are $\rho$-regular for a reasonable constant. The most regular distribution is the uniform distribution itself. Here, our analysis provides the biggest improvements. We prove that even in this case CATE estimation remains hard, requiring $\Omega(M/\epsilon^2)$ samples. This formally proves a quadratic separation between treatment allocation and CATE estimation. In particular, CATE estimation remains hard even after imposing $\rho$-regularity.

**Empirical evaluation.** We consider data from five different randomized control trials (RCTs). For each dataset, we evaluate how many samples are needed to obtain a near-optimal allocation. To do so, we treat the treatment effects estimated from the full RCT dataset as the ground truth. We then subsample the dataset to various sample sizes and compute the allocation value realized

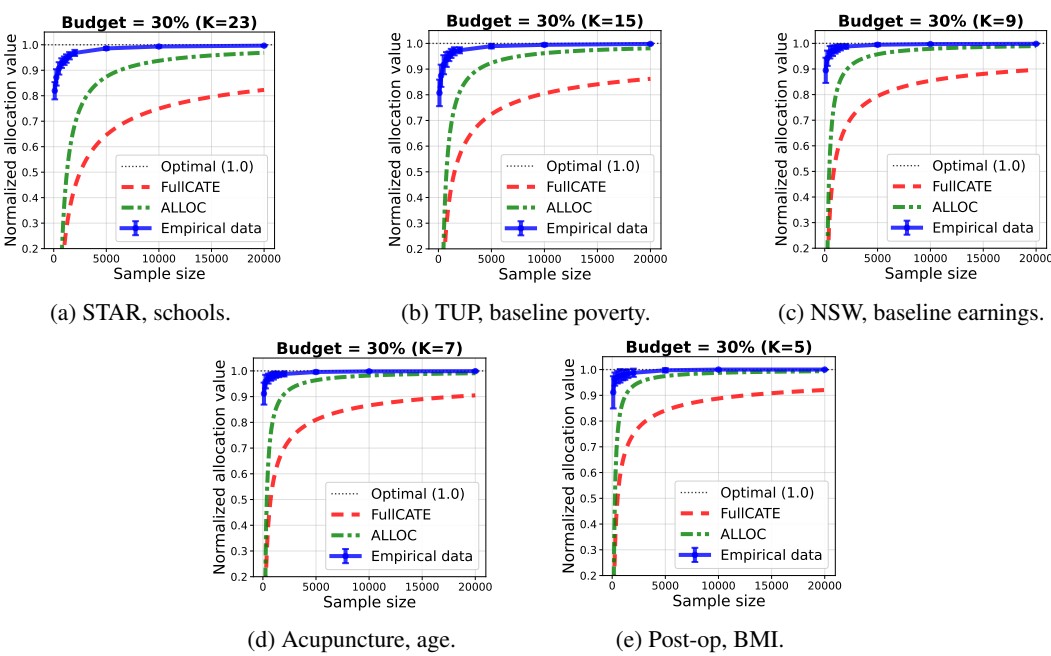

Figure 1: In all cases, we realize a close-to-optimal allocation with very few samples (blue), much less than the worst-case of $O(M/\epsilon^2)$ (red) and even less than our theoretical bound $O(M/\epsilon)$ (green).

by our algorithm. As shown in Figure 1, in all cases we find near-optimal allocations with fewer samples than our theoretical upper bound predicts. This suggests that real-world treatment effect distributions meet the assumptions of our theory. As a robustness check, we replicate the results across various different partitions of the populations and for varying budget sizes.

**Discussion and limitations.** There's an immediate practical takeaway relevant to future policy decisions about targeting welfare-promoting interventions. The standard sample size calculations for CATE estimation are excessively pessimistic for the purpose of treatment allocation. Indeed, we can find nearly optimal allocations from coarse treatment effect estimates. Perhaps counterintuitively, an RCT that is severely underpowered for CATE estimation can still yield excellent allocations. As a rule of thumb, $M/\epsilon$ samples suffice for a $(1-\epsilon)$-optimal allocation.

While our smoothness requirement is typically met in practice, it may not always hold. We address this limitation in two ways. First, we expose an exact instance-dependent optimality condition (Section 4.2) that can be computed from coarse estimates as well. Thus, a policy maker can *certify* from few samples that a solution is near-optimal or—if it isn't—invest in additional samples. Alternatively, we consider strategies for the policy maker to mitigate cases where there are too many units around the optimal threshold $\tau_K$ *without* additional samples. Specifically, we study the strategies of underspending and overspending on the original budget $K$ in order to obtain a $(1-\epsilon)$-optimal allocation with $O(M/\epsilon)$ many samples. We show that, in practice, in the few cases where a close-to-optimal allocation is not realized, we can find a very close threshold $\tau_{K'}$ at which we do realize it with only $O(M/\epsilon)$ many samples. To summarize, our work strongly separates the sample complexity treatment effect *estimation* from that of treatment *allocation*. The latter requires far fewer samples than the former. This stark separation has the potential to inform future policy decisions.

## 1.2 RELATED WORK

Perdomo et al. (2025) and Shirali et al. (2024) inspired our work by showing that effective resource allocation need not require accurate predictions. These works bring into focus a critical examination of prediction in resource allocation (Barabas et al., 2018; Perdomo, 2024; Shirali et al., 2025; Fischer-Abaigar et al., 2025; Mashiat et al., 2025). Our results extend this line of work with a fundamental observation: Extremely coarse treatment effect estimates can still yield near-optimal

allocations. Perhaps closest to our work, Shirali et al. (2024) contrast unit-level allocations with individual-level targeting, showing that a measure of inter-unit inequality makes unit-level allocations competitive with individual-level targeting. In our work, allocations are always unit-level; we study how accurate unit-level treatment effect estimates have to be for the purpose of allocation.

Another recent line of work studies different strategies for using RCT data effectively (Sharma & Wilder, 2025; Cortes-Gomez et al., 2024; Wilder & Welle, 2025). In line with these results, our work provides further evidence that investing in greater accuracy is not always necessary in order to achieve a more efficient intervention. Unlike these works, our paper focuses on sample complexity upper bounds for treatment allocation, rather than on considering alternative strategies for estimating treatment effects. There is a vast causal inference literature on learning optimal targeting rules that address heterogeneity in the population (Manski, 2004; Qian & Murphy, 2011; Kitagawa & Tetenov, 2018; Kallus, 2018; Athey & Wager, 2021). Much work tackles the case of large $M$ using machine learning methods (Chernozhukov et al., 2018; Nie & Wager, 2021), which is not our focus. There's also much work targeting rules subject to budget constraints, e.g., Bhattacharya & Dupas (2012); Luedtke & Van Der Laan (2016); Le et al. (2019). Our work also relates to the literature on bandits, specifically to the problem of identifying good arms (Audibert & Bubeck, 2010) and the numerous variants of this problem, such as identifying any subset of a set of good arms (Kalyanakrishnan et al., 2012; Bubeck et al., 2013; Chaudhuri & Kalyanakrishnan, 2019; Rejwan & Mansour, 2020). In our setting, however, we are not interested in an adaptive procedure; we want our algorithm to be implementable in practice. The lower bound for CATE estimation runs through the bandit literature in various guises, see, e.g., Kaufmann et al. (2016); Katz-Samuels & Jamieson (2020).

## 2 NOTATION AND PRELIMINARIES

We have a population $\mathcal{X}$ divided into a set $\mathcal{U}$ of $M$ units. Each unit $u \in \mathcal{U}$ is independent from the others, and we wish to select $K$ out of the $M$ units to carry out an intervention. We operate in the setting of a resource-constrained positive intervention: ideally, we would like to treat all units, but a budget limits the number of units that we can intervene on. In order to decide which units to select, a typical approach is for the policy-maker to estimate the average treatment effect $\tau(u)$ in each unit $u$, and then choose the $K$ units with the highest $\tau(u)$ values. Throughout we assume bounded treatment effects $\tau(u) \in [0, 1]$, normalized to the unit interval. Due to space constraints, we defer all proofs of the main body to the appendix, while providing some proof sketches.

Estimating the average treatment effect within a unit—either via an RCT or an observational design—is the well-established problem of causal inference that is not the subject of this work. We therefore assume that we can get the average treatment effect $\tau(u)$ up to an additive error $\epsilon > 0$ at the cost of $O(1/\epsilon^2)$ samples. This assumption hides the challenges of causal inference, but exposes the sample complexity relevant for our argument.

**Definition 1.** Call $\hat{\tau}$ an $(\epsilon, \delta)$-accurate estimate *of $\tau$ if $|\hat{\tau} - \tau| \leq \epsilon$ with probability $1 - \delta$.*

**Definition 2** (Estimation oracle). *Given a unit $u \in \mathcal{U}$ and parameters $\epsilon, \delta > 0$, an* estimation oracle $\mathcal{O}$ *returns an $(\epsilon, \delta)$-accurate estimate $\hat{\tau}(u)$ of $\tau(u)$ at the cost of $O(\ln(2/\delta)/\epsilon^2)$ samples.*

Throughout the paper, we always assume that the estimation oracle $\mathcal{O}$ is available to the algorithm. The sample complexity stated in Definition 2 follows from Hoeffding's inequality. The upper bound is tight up to constants except in special cases. This follows from standard lower bounds for mean estimation in Bernoulli families (Le Cam, 1973; Wasserman, 2004; Tsybakov, 2008). These lower bounds extend to estimating average treatment effects (Imbens & Rubin, 2015). Typically, in the setting of treatment effect estimation, we compute an $(\epsilon, \delta)$-accurate estimate for every unit. We refer to this problem as the FullCATE problem.

**Definition 3** (FullCATE problem). *Given a population $\mathcal{X}$ of individuals divided into a set of $M$ units $\mathcal{U}$, solving the FullCATE$(\mathcal{X}, \mathcal{U}, \epsilon, \delta)$ problem consists of producing $M$ estimates $\hat{\tau}$ such that $\hat{\tau}(u)$ is an $(\epsilon, \delta)$-accurate estimate of $\tau(u)$ for each $u \in \mathcal{U}$.*

We can compute the sample complexity of the FullCATE problem if we solve it by calling the estimation oracle with parameters $(\epsilon, \delta)$ for each unit $u \in \mathcal{U}$, making a total of $M$ independent calls. This upper bound is again tight, in general, up to constant factors when groups are non-overlapping, since each sample can only contribute to estimating one of the treatment effects.

**Lemma 2.** *Having access to an estimation oracle $\mathcal{O}$, we can solve the* FullCATE$(\mathcal{X}, \mathcal{U}, \epsilon, \delta)$ *problem with a total of $N_{\mathsf{FullCATE}} = O(M \ln(2/\delta)/\epsilon^2)$ samples from $\mathcal{X}$.*

## 2.1 Allocation versus Estimation

Our goal is to find a near-optimal allocation without incurring the cost of solving FullCATE. This means selecting the $K \leq M$ units with the highest true treatment effect values $\tau(u)$, where $K$ is determined by the given budget. We think of the treatment as a positive intervention, which is why we focus on identifying the units with the *highest* values of $\tau(u)$. For an integer $K \in \{1, \ldots, M\}$, we denote the $K$-th largest value of $\tau(u)$ among the $M$ possible treatment effect values by $\tau_K$.

**Definition 4** (Value)**.** *Given a set $\mathcal{U}$ of $M$ units and a budget $K \leq M$, an* allocation function *$g$ returns a set $\mathcal{U}_g \subseteq \mathcal{U}$ of $K$ units. The* value *of the allocation function over an interval $A \subseteq [0, 1]$, denoted $V_{\mathcal{U}_g}(A)$, is defined as $V_{\mathcal{U}_g}(A) = \sum_{u \in \mathcal{U}_g | \tau(u) \in A} \tau(u)$. The (total) value of the allocation function $g$ for budget $K$ is equal to $V_{\mathcal{U}_g}([0, 1])$.*

We abbreviate $V_{\mathcal{U}_g}([0, 1])$ by $V_{\mathcal{U}_g}$ if the interval $A = [0, 1]$ can be inferred from the context. The optimal allocation for a given budget is the one that realizes the highest value.

**Definition 5.** *Given a budget to treat $K$ units, the optimal allocation function is the one that selects the $K$ values $\tau(u)$ that maximize the value $\sum_{u \in \mathcal{U}} \tau(u)$ over all subsets $\mathcal{U}$ of $K$ units. We denote the set of units chosen by the optimal allocation function by $\mathcal{U}^*$.*

Throughout, we assume that the true treatment effect values $\tau(u)$ are unique (otherwise, we can slightly perturb them), and thus the optimal allocation is unique.

**Definition 6** (ALLOC problem)**.** *Given a budget $K \leq M$ and parameters $\epsilon, \delta$, we say that a set $\mathcal{U}_g \subseteq \mathcal{U}$ of $K$ units is an $(\epsilon, \delta)$-optimal allocation* ALLOC$^*$ *if $\frac{V_{\mathcal{U}_g}([0,1])}{V_{\mathcal{U}^*}([0,1])} \geq 1 - \epsilon$ with probability at least $1 - \delta$, where the probability is taken over the coins of $g$. Solving the* ALLOC$(\mathcal{X}, \mathcal{U}, K, \epsilon, \delta)$ *problem consists of selecting a set $\mathcal{U}_g$ of $K$ units that is a $(1 - \epsilon, \delta)$-optimal allocation.*

We usually drop the parameter $\delta$, implying that the guarantee holds with high probability. Crucially, the value of an allocation is computed with the true $\tau$ values rather than with the estimated $\hat{\tau}$ values. The standard way of solving the ALLOC problem is through the FullCATE problem.

**Lemma 3.** *Given any budget $K \leq M$ and parameters $\epsilon, \delta$, we can solve the* ALLOC$(\mathcal{X}, \mathcal{U}, \epsilon, \delta)$ *problem with $N = O(M \ln(2/\delta)/\epsilon^2)$ samples from $\mathcal{X}$.*

## 3 Directly targeting units for allocation

Recall, the optimal allocation for a given budget $K$ chooses the $K$ units with the largest treatment effect value $\tau(u)$. It is clear that $\mathcal{U}^* = \{u \mid \tau(u) \geq \tau_K\}$ (Claim B.1). Hence, solving the allocation problem reduces to solving a threshold problem, where for a given budget $K$, the task is to determine whether $\tau(u) \geq \tau_K$ or $\tau(u) < \tau_K$ for each $u \in \mathcal{U}$. In the former case, we decide to intervene on unit $u$; otherwise we do not. In order to solve this thresholding problem, we do not need to estimate each of the $\tau(u)$ values up to accuracy $\epsilon$; we only need to determine whether $\tau(u)$ lies above or below $\tau_K$. The farther $\tau(u)$ is from $\tau_K$, the less accuracy we need for our corresponding $\hat{\tau}(u)$ estimate. Conversely, we only need high accuracy in estimating $\hat{\tau}(u)$ when $\tau(u)$ is close to the threshold $\tau_K$.

## 3.1 Low-accuracy estimation algorithm

We introduce another parameter $\rho$ that quantifies the error to which we estimate each treatment effect. Intuitively, we only try to determine whether $\tau(u)$ is above $\tau_K + 2\rho$ or not. If we believe that $\tau(u)$ belongs to the interval $[\tau_K, \tau_K + 2\rho]$, then we "give up" and stop trying to determine the precise value of $\tau(u)$. As we show, the value of $\rho$ that strikes the right balance between a low sample complexity and a close-to-optimal allocation value is $\rho = \Theta(\sqrt{\epsilon})$. That is, instead of estimating all treatment effect values up to accuracy $\epsilon$, we only estimate them up to accuracy of the order $\sqrt{\epsilon}$ and then select the top $K$ values based on these coarse estimates. This yields the following low-accuracy algorithm (Algorithm 1), which is non-adaptive so that it can be implemented in a typical one-round RCT (further sample complexity reductions can be achieved by introducing adaptivity, for example with a multiple stage RCT, but these are not usually implementable in practice).

---

**Algorithm 1** Low-estimation non-adaptive allocation (LEA)

    **Input:** $M$ units, budget $K$, parameters $\epsilon, \delta, \gamma > 0$, where $\gamma = \Theta(1)$.

    For each unit $u$, obtain a $(\rho, \delta/M)$-accurate estimate $\hat{\tau}(u)$ for $\rho = \gamma\sqrt{\epsilon}$.

    **Output:** A set $\mathcal{U}_{\mathsf{LEA}}$ consisting of the top $K$ units in sorted order of $\hat{\tau}(u)$ values.

---

By definition, the algorithm selects all units $u$ such that $\hat{\tau}(u) \geq \hat{\tau}_K$. First, by Hoeffding's bound and the choice of $\rho$, the algorithm requires a number of samples proportional to $1/\epsilon$, rather than $1/\epsilon^2$.

**Lemma 4.** *Given any $\mathcal{X}, \mathcal{U}, K$, Algorithm 1 requires $O(M \ln(2M/\delta)/\epsilon)$ many samples from $\mathcal{X}$.*

By a union bound, all units $u$ satisfy $|\hat{\tau}(u) - \tau(u)| \leq \rho$ with probabbility at least $1 - \delta$. Through a careful analysis of the estimates, we also show the following:

**Lemma 5.** *The output estimates $\hat{\tau}_K$ and the set $\mathcal{U}_{\mathsf{LEA}}$ returned by Algorithm 1 satisfy the following properties: (1) $|\hat{\tau}_K - \tau_K| \leq \rho$. (2) All units $u$ such that $\tau(u) > \tau_K + 2\rho$ belong to $\mathcal{U}_{\mathsf{LEA}}$. (3) All units $u$ such that $\tau(u) < \tau_K - 2\rho$ do not belong to $\mathcal{U}_{\mathsf{LEA}}$.*

In other words, our algorithm selects the units optimally in the intervals $(\tau_K + 2\rho, 1]$ and $[0, \tau_K - 2\rho)$. We only need to worry about the interval surrounding $\tau_K$. This motivates the following definitions:

**Definition 7.** *We define the intervals $A_1 = (\tau_K + 2\rho, 1]$ and $D = [\tau_K, \tau_K + 2\rho]$. We also let $\mathcal{U}^0_{\mathsf{LEA}} = \{c \in \mathcal{U}_{\mathsf{LEA}} \mid \tau(u) \notin A_1\}$, $K_1 = |\{u \mid \tau(u) \in A_1\}|$, and $K_0 = K - K_1$.*

The guarantees of Lemma 5 imply that $V_{\mathcal{U}_{\mathsf{LEA}}}(A_1) = V_{\mathcal{U}^*}(A_1)$, and moreover allow us to lower bound $V_{\mathcal{U}^0_{\mathsf{LEA}}}([0, 1])$ with $(\tau_K - 2\rho)K_0$, and upper bound $V_{\mathcal{U}^*}(D)$ with $(\tau_K + 2\rho)K_0$. This allows us to get the following bound, where we use $V(A_1)$ as a short-hand for $V_{\mathcal{U}_{\mathsf{LEA}}}(A_1) = V_{\mathcal{U}^*}(A_1)$.

**Claim 3.1.** $\dfrac{V_{\mathcal{U}_{\mathsf{LEA}}}([0,1])}{V_{\mathcal{U}^*}([0,1])} \geq 1 - \dfrac{4\rho K_0}{V(A_1) + (\tau_K + 2\rho)K_0}.$

We let $\Pr_\tau$ denote the discrete distribution of $\tau(u)$ values. The quantity $\Pr_\tau[\tau_K, \tau_K + 2\rho]$ and its relationship to $2\rho M$ is key to our analysis. To that end, we let $\Pr_{\tau_K}[\tau_K, \tau_K + 2\rho] = \theta_K$ and $V(A_1) = \gamma_1 M$. This leads to the following general accuracy bound for Algorithm 1:

**Claim 3.2.** *Given any $\mathcal{X}, \mathcal{U}, K$, Algorithm 1 returns a $\left(1 - \dfrac{4\gamma\theta_K}{\gamma_1 + \tau_K\theta_K} \cdot \sqrt{\epsilon}\right)$-optimal approximation with $O(M \ln(2M/\delta)/\epsilon)$ many samples.*

In Section 4, we show how the expression in Claim 3.2 simplifies to $(1 - \epsilon)$ in the case of $\rho$-regular distributions, thus attaining a $(1 - \epsilon)$-OPT allocation.

## 3.2 QUANTILES SUFFICE

A key insight in our analysis is the fact that for the problem of treatment allocation we only need to know quantiles of the distribution. In other words, it's enough to approximate the CDF of the distribution $\Pr_\tau$ of treatment effect values.

**Definition 8.** *Let $F(t) = \Pr_\tau[\tau(u) \leq t]$ be the CDF of the distribution of $\tau(u)$ values, and let $\hat{F}$ denote the CDF corresponding to the distribution $\Pr_{\hat{\tau}}$ of the low-accuracy estimates $\hat{\tau}(u)$ produced by Algorithm 1. That is, $\hat{F}(t) = \frac{1}{M}|\{u : \hat{\tau}(u) \leq t\}|$.*

We denote the PDF by $f(t)$. Given a budget of $K \leq M$, $\tau_K$ definitionally corresponds to the value in $[0, 1]$ such that $F(\tau_K) = 1 - K/M$; i.e., the $K$-th $M$-th quantile of the distribution of $\tau$ values. Similarly, $\gamma_1$ and $V_{\mathcal{U}^*}$ can be computed as permutation-invariant quantities derived from the CDF:

**Claim 3.3.** *For any budget $K \leq M$, we can compute $\tau_K, \gamma_1, V_{\mathcal{U}^*}$ as: (1) $\tau_K = F^{-1}(1 - K/M)$, (2) $\gamma_1 = \frac{1}{M} \int_{\tau_K + 2\rho}^{1} t \, dF(t)$, (3) $V_{\mathcal{U}^*}([0, 1]) = \int_{\tau_K}^{1} t \, dF(t)$.*

Moreover, the $\rho$-accurate estimates $\hat{\tau}(u)$ naturally provide an approximation of $f$ and $F$, which we can in turn use to check, from the low-accuracy estimates $\hat{\tau}_K$, the smoothness behavior of the treatment effect values around the optimal threshold:

**Claim 3.4.** *For any $t \in [0, 1]$, $\hat{F}(t - \rho) \leq F(t) \leq \hat{F}(t + \rho)$.*

## 4 THE SAMPLE COMPLEXITY OF TREATMENT ALLOCATION

To illustrate the key ideas, we show that if the treatment effect values $\tau(u)$ are uniformly distributed, our allocation algorithm achieves a $(1 - \epsilon)$-optimal allocation with $O(M/\epsilon)$ many samples. At the same time, any estimator that computes all $M$ treatment effect values values $\tau(u)$, each within $\epsilon$-accuracy, must use at least $\Omega(M/\epsilon^2)$ many samples. Therefore, the uniform distribution demonstrates a sample complexity gap between the problem of obtaining an optimal allocation and the problem of full CATE estimation. By *uniformly distributed* we mean that we place the $\tau(u)$ values uniformly-spaced in $[0, 1]$ and randomly permute them.

**Theorem 6.** *Let the treatment effect values be uniformly distributed. Then: (1) We can obtain a $(1 - \epsilon, \delta)$-approximation of the optimal allocation with $N = O(M \ln(2M/\delta)/\epsilon)$ many samples from $\mathcal{X}$ for any budget $K \leq M$. (2) FullCATE requires $\Omega(M/\epsilon^2)$ samples.*

*Proof sketch.* The sample complexity bound for Part (1) follows from Lemma 4. The $\epsilon$-optimality of the realized allocation value follows as a consequence of Claim 3.1 using the fact that $\theta_K = \Pr_\tau[\tau_K, \tau_K + 2\rho] \approx 2\rho$, given that the $\tau(u)$ values are uniformly distributed by assumption. This allows us to obtain the term $\rho^2$ in the numerator of the expression $4\rho\theta_K/(\gamma_1 + (\tau_K + 2\rho)\theta_K)$, which is the key to achieving a $(1 - (\sqrt{\gamma}\rho)^2) = (1 - \epsilon)$-optimal allocation. The lower bound follows from a minimax argument based on LeCam's Theorem.

### 4.1 SMOOTH DISTRIBUTIONS

We can generalize our $O(M/\epsilon)$ upper bound beyond the uniform distribution. All we require is for the $D$-interval to have "uniform-like" mass, so that $\theta_K \approx 2\rho$ and thus our proof of Theorem 6 can still go through (i.e., to obtain the $\rho^2$ term in the numerator). Because $\tau_K$ could be anywhere, we require this condition on all intervals that have width at least $2\rho$, a condition that we call *$\rho$-regularity*:

**Definition 9.** *Given $\rho > 0$, we say that a distribution $Z$ on $[0, 1]$ is $\rho$-regular if for all intervals $S \subseteq [0, 1]$ such that $|S| \geq 2\rho$ there exists a constant $c = \Theta(1)$ satisfying $\Pr_Z[S] \leq c \cdot \mathcal{U}[S]$, where $\mathcal{U} = \Pr_\mathcal{U}$ and $\mathcal{U}$ denotes the uniform distribution over $[0, 1]$.*

We can view this condition as a form of smoothness requirement. Note that this is a much looser notion than requiring, for example, $O(\rho)$ total variation distance between $Z$ and $\mathcal{U}$, or with any other distance measure to the uniform that is not permutation invariant. Note that the value of $c$ is always upper-bounded by the supremum of the density of the distribution $Z$; i.e., $c \leq \sup_t f(t)$.

**Theorem 7.** *Let the treatment effect values be distributed according to a $\rho$-regular distribution for $\rho = \Theta(\sqrt{\epsilon})$. Then, we can obtain a $(1 - \epsilon, \delta)$-approximation of the optimal allocation with $N = O(M \ln(2M/\delta)/\epsilon)$ many samples from $\mathcal{X}$ for any budget $0 \leq K \leq M$.*

*Proof sketch.* The $\rho$-regularity condition ensures that $\theta_K = \Pr_\tau[\tau_K, \tau_K + 2\rho] = 2\rho c$ for $c = \Theta(1)$. Following Claim 3.1, this allows us to again obtain the term $\rho^2$ in the numerator of the expression $4\rho\theta_K/(\gamma_1 + (\tau_K + 2\rho)\theta_K)$, thus ensuring that the ratio is upper bounded by $\epsilon$.

Note that for *any* distribution we can get a $(1 - \epsilon)$-OPT approximation by setting $\gamma = \sqrt{\gamma_1/8c}$ when calling Algorithm 1, by letting $c$ be the minimum value such that $\Pr_Y[S] \leq c \cdot \mathcal{U}[S]$ for all intervals $S$ such that $|S| \geq 2\rho$. However, if $c$ is not $\Theta(1)$, then $\rho$ might not be of the order $\Theta(\sqrt{\epsilon})$, and so the sample complexity will no longer be of the order $O(M/\epsilon)$. That is, for an arbitrary $\Pr_\tau$:

**Theorem 8.** *Let $c$ be the minimum value such that $\Pr_\tau[S] \leq c \cdot \mathcal{U}[S]$ for all subsets $S \subseteq [0, 1]$ such that $|S| \geq 2\rho$. For any budget $K \leq M$, we call Algorithm 1 with with $\rho = \sqrt{V_{\mathcal{U}^*}/8c}$. This gives a $(1 - \epsilon)$-OPT approximation of the optimal allocation with $O(M \ln(2M/\delta)/\rho^2)$ many samples.*

We remark that $V_{\mathcal{U}^*}$ is short-hand for $V_{\mathcal{U}^*}([0, 1])$ and that it is a function of $K$. If the distribution is $\rho$-regular, then $c = \Theta(1)$, in which case we require $O(M/\epsilon)$ many samples, recovering Theorem 7. Note that solving the FullCATE problem to accuracy $\epsilon$ already required having a bound on $\tau_K$ (Claim 3), if we want to ensure exactly that we get a $(1 - \epsilon)$-optimal allocation. Similarly, here we use an upper bound of $\gamma$ in order to ensure exact $\epsilon$-optimality.

*Why $\rho$-regularity is not a demanding property.* As discussed, $\rho$-regularity is a permutation-invariant property that only depends on the shape of the CDF. It also ensures that the threshold $\tau_K$ that determines which units are selected for treatment does not yield a highly arbitrary allocation. Empirically,

as we extensively show in our experiments with real-world RCT data in Section 6, our algorithm highly succeeds in practice, further demonstrating that $\rho$-regularity is a natural condition. Moreover, typical families of distributions of treatment effect values are $\rho$-regular for reasonable values of $\gamma$. As we compute in Section C.1 using the CDF-derived quantities shown in Claim 3.3, we can explicitly compute the values of $\tau_K$, $c$, $V_{\mathcal{U}^*}$, and $\gamma$ for typical distributions of treatment effect values from their CDF, which we do for the uniform, Beta, and Gaussian distribution families. In all of these cases, $\gamma$ ranges between 0.07 and 0.2.

## 4.2 GENERAL OPTIMALITY CONDITION

Theorem 7 is not an if and only if statement, and so $\rho$-regularity is a sufficient but not necessary condition for Algorithm 1 to obtain a near optimal allocation. This means that, in some cases, we can obtain optimal allocations even if there is a lot of mass in the interval around the optimal threshold $\tau_K$ (e.g., if the units are extremely close to $\tau_K$). We conclude this section by providing a necessary and sufficient condition on the CDF of the distribution of $\tau$ values (for the worst-case high probability instance of Algorithm 1), thus providing a precise instance-dependent upper bound on the sample complexity of near-optimal treatment allocation. To do so, we require the following definition:

**Definition 10.** *Given budget* $K \leq M$, $\alpha_K \in [0, 1]$ *corresponds to the smallest value such that* $\Pr_\tau \left[ [\tau_K - 2\rho, \tau_K - \alpha_K] \right] = \Pr_\tau \left[ [\tau_K, \tau_K + 2\rho] \right].$

**Claim 4.1.** *For any budget* $K \leq M$, *Algorithm 1 called with parameters* $\epsilon, \delta, \rho$ *returns a* $(1 - \epsilon, \delta)$-OPT *allocation if and only if the distribution of treatment effect values satisfies:*

$$\int_{\tau_K}^{\tau_K + 2\rho} t f(t)\, dt - \int_{\tau_K - 2\rho}^{\tau_K - \alpha_K} t f(t)\, dt\, dt \leq \epsilon \int_{\tau_K}^{1} t f(t)\,. \tag{1}$$

As we showed in Section 3.2, our estimation algorithm provides an approximation to the CDF of the distribution of treatment effect values. Specifically, we can use it as a certificate to check whether Equation 1 is satisfied (note that this is only a one-sided guarantee, given that we use various upper bounds). To do so, we compute an upper bound of the expression $(V_{\mathcal{U}^*}(D) - V_{\mathcal{U}^0_{\mathsf{LEA}}})/V_{\mathcal{U}^*}([0, 1])$ using our $\hat{\tau}$ estimates. We do so in Section C.2 by using Claim 3.4. In the same section, we give an example of Claim 6 in the case where the treatment effect values are uniformly distributed in $[0, 1]$.

## 5 FLEXIBLE BUDGET

When the original $\tau_K$ threshold has too many units around it for our algorithm to return an optimal allocation, we can slide the budget $K$ up or down slightly into a new budget $K'$ so as to find a new threshold $\tau_{K'}$ that lies in an interval of low probability mass (where $\rho$-regularity holds). In our experiments in Section 6, we find the closest $K'$ to $K$ that obtains a $(1 - \epsilon)$-optimal allocation with only $O(M/\epsilon)$ samples, in the few cases where the original $K$ does not give an optimal allocation.

*Very dense distributions.* Such a budget $K'$ does not always exist: e.g., if all $\tau(u)$ values are the same, then the interval $[\tau_K, \tau_K + 2\rho]$ does not satisfy $\rho$-regularity for any $K \in [1, M]$. However, in this case, Algorithm 1 always returns an optimal allocation, since any selection of $K$ units yields the optimal $K \cdot \tau(u)$ value (this is also shown by Claim 4.1, since $V_{\mathcal{U}^*}(D) - V_{\mathcal{U}^0_{\mathsf{LEA}}} = 0$). In Section D we provide a "2 spikes" example where we show that no budget is able to achieve an optimal allocation. Nonetheless, we show that we *almost* achieve an optimal allocation, and so in order to realize the required value, we can either promise a $(1 - \kappa\epsilon)$-OPT allocation for a small $\kappa$, or we can select more units for intervention. This type of resource augmentation is different from the "sliding budget" approach discussed at the beginning of this section. Here, we add more units to our allocation $\mathcal{U}_{\mathsf{LEA}}$, but we compete against the *original* budget $K$. For example, as we show in Section D for the 2 spikes case, increasing the budget to $K' = K(1 + 3\epsilon)$ ensures the $\epsilon$-optimality of a random allocation. In the appendix, we apply this overspending idea to distributions that are *not* $\rho$-regular around the initial budget threshold $\tau_K$. As we show, adding an extra $\approx \sqrt{\epsilon} K_0$ units suffices to obtain a $(1 - \epsilon)$-optimal allocation. In our empirical results in Section 6, we obtain that in practice we only require adding one extra unit in order to realize the value of the optimal allocation.

*Choosing an appropriate budget.* In Section D, we discuss various strategies for when the policy maker has some flexibility over the budget, where it is sensible to refine the number of units that

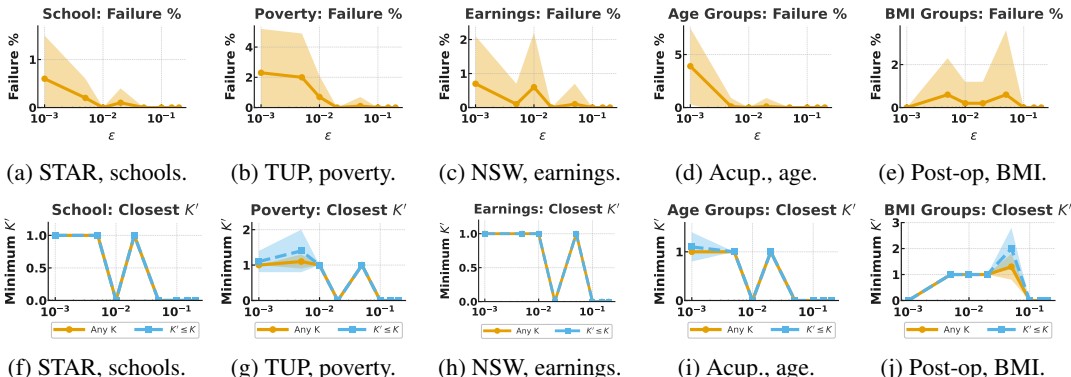

Figure 2: Top row: failure rate vs $\epsilon$. Bottom row: closest $K'$ value vs $\epsilon$. Each row represents one dataset (STAR, TUP, NSW, Acup., Post-op) with one of the unit grouping methods that we study.

are selected for intervention after running Algorithm 1. These include: Detecting when a budget does not work by using the approximation $\hat{F}$ of the CDF (this is a one-sided test, as per the test of Claim 1), sliding the budget from $K$ to $K'$ as to find a $\rho$-regular interval, and adding a few extra units to $\mathcal{U}_{\text{LEA}}$ as to realize the optimal value (overspending), in the cases where the distribution is dense throughout a large range. We test the flexible budget strategies experimentally in Section 6.

## 6 EXPERIMENTS WITH REAL-WORLD RCTs

We test our allocation algorithm extensively on five different real-world RCT datasets across different domains, these set of five datasets was recently studied in Sharma & Wilder (2025). These are: the (1) *Tennessee's Student Teacher Achievement Ratio (STAR) project* (Pate-Bain et al., 1997) (eduction), (2) *Targeting the Ultra Poor (TUP) in India* (Banerjee et al., 2021) (economic development), (3) *National Supported Work (NSW) demonstration* (Dehejia & Wahba, 2002; 1999; LaLonde, 1986) (labor economics), (4) *Acupuncture dataset* (Vickers et al., 2004) (healthcare), and (5) *Postoperative Pain dataset* (McHardy & Chung, 1999) (healthcare). We describe these in detail, as well as our experimental set-up, in Section E and Section F.

For each dataset, we group individuals in different ways using the covariates, ensuring that each group is somewhat balanced in terms of control vs treatment. Each different grouping method yields a partition $\mathcal{U}$ of $M$ units. For each unit $u$, we define $\tau(u)$ to be the average treatment effect within the unit across the entire dataset. Then, we run our semi-synthetic experiments as follows: every time we sample from the population, we choose a unit $u$ uniformly at random, and then receive a sample i.i.d. from $\text{Bern}(\tau(u))$. This gives us estimates $\hat{\tau}(u)$ for each $u$. This semi-synthetic setting allows us to know the ground-truth of the value realized by the optimal allocation. We repeat the whole sampling procedure 50 times, to be able to obtain confidence bounds. For the plots in Figure 1, for various sample sizes, we compute the corresponding estimates $\hat{\tau}$, and then the realized allocation value for various budgets $K \leq M$. For the plots in Figure 2, we repeat the sampling simulation for different fixed values of $\epsilon$ (and in turn repeating each simulation 50 times). For each, we call Algorithm 1 and obtain a set of units $\mathcal{U}_{\text{LEA}}$ for each budget $K$. For *all* budgets $K \in [1, M]$, we check whether or not $\mathcal{U}_{\text{LEA}}$ is an $(1 - \epsilon)$-optimal allocation, and compute the percentage of budgets (out of a total of $M$ possible budgets) that fail to be $(1 - \epsilon)$-optimal (top row in Figure 1). For the budgets that fail, we compute the closest budget $K'$ that yields an $(1 - \epsilon)$-optimal allocation (blue line, bottom row in Figure 1), and the closest budget $K'$ satisfying $K' \leq K$; i.e., if we only consider underspending (orange line, bottom row in Figure 1). We see that, on average, the failure rate always says below 5%, and that $|K' - K|$ is typically 1, even if we are only allowed to underspend ($K' \leq K$). We also try the resource augmentation approach of adding more units, and in all failed budget cases we always realize a $(1 - \epsilon)$-optimal value by adding just one extra unit.

We provide all the details of the experiments and datasets along with the rest of figures (which include all of our grouping methods for each dataset, thus replicating Figures 1 and 2 extensively across all datasets and groupings) in the appendix in Sections E and F.

## ACKNOWLEDGMENTS

We thank Rediet Abebe, Florian Dorner, Kevin Jamieson, Juan Carlos Perdomo, Ali Shirali, and Bryan Wilder for helpful discussions, pointers, and feedback.

## ETHICS STATEMENT

As we argue in this paper, treatment allocation is a problem of societal interest, given that policy makers use allocation algorithms and run randomized control trials in order to decide which groups of the population to intervene on. Studying algorithms for such interventions can thus crucially affect how these are performed in practice. In our paper, we only use publicly-available data from RCTs that have already been conducted, and so our study does not perform any real-world intervention. Still, we hope that our results can be beneficial in guiding treatment allocations in practice; specifically, our results suggest that good allocations can be attained with far fewer samples than the usual CATE estimation bound would suggest, and thus should be much cheaper and easier to implement in practice.

## REPRODUCIBILITY STATEMENT

We provide all proofs for our mathematical statements in the appendix. We provide a detailed summary of the set-up of our experiments in Section E, and we provide the code that we have used to generate all of our plots and results in the supplementary material package for each of the five datasets. All the RCT datasets that we explore in this paper are publicly available on the Internet.

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

## A  DEFERRED PROOFS FROM SECTION 2

**Lemma 9.** *We can compute an $(\epsilon, \delta)$-accurate estimate $\hat{\tau}(u)$ of $u$ with $O(\ln(2/\delta)/\epsilon^2)$ samples.*

*Proof.* By Hoeffding's inequality, given that $\tau(u) \in [0, 1]$ for all $u$, it follows that with probability at least $1 - \delta$,

$$\Pr\left[\left|\hat{\tau}(u) - \tau(u)\right| \geq \epsilon\right] \leq 2e^{-2m\epsilon^2}.$$

Hence, if we want $\hat{\tau}(u)$ to be an $(\epsilon, \delta)$-accurate estimate of the true value $\tau(u)$, we need to take $m \geq O(\ln(2/\delta)/\epsilon^2)$ many samples from $u$. $\qquad\square$

**Lemma 3.** *Given any budget $K \leq M$ and parameters $\epsilon, \delta$, we can solve the $\mathsf{ALLOC}(\mathcal{X}, \mathcal{U}, \epsilon, \delta)$ problem with $N = O(M \ln(2/\delta)/\epsilon^2)$ samples from $\mathcal{X}$.*

*Proof.* By Lemma 2, we can solve the $\mathsf{FullCATE}(\mathcal{X}, \mathcal{U}, \epsilon, \delta)$ problem with parameters $\mathcal{X}, \mathcal{U}, \epsilon, \delta$ with $O(m \ln(2/\delta)/\epsilon^2)$ samples from $\mathcal{X}$. We use parameters $\epsilon'$ and $\delta/M$, for $\epsilon'$ to be determined in the proof. This produces $M$ estimates $\hat{\tau}$ such that $\hat{\tau}(u)$ is an $(\epsilon', \delta/N)$-accurate estimate of $\tau(u)$ for each $u \in \mathcal{U}$. We then define the following allocation function $g$: select the $K$ units with the highest estimated values $\hat{\tau}(u)$. This defines the set of units $\mathcal{U}_g$.

By definition of $\tau_K$, it follows that $V_{\mathcal{U}^*}([0, 1]) \geq K\tau_K$. In the worst case, all the units that we select are wrong, and by the $\epsilon'$ accuracy guarantee we have that, in general, $V_{\mathcal{U}^*}([0, 1]) - V_{\mathcal{U}_g}([0, 1]) \leq 2K\epsilon'$, and equivalently

$$\frac{V_{\mathcal{U}_g}([0, 1])}{V_{\mathcal{U}^*}([0, 1])} \geq 1 - \frac{2K\epsilon'}{V_{\mathcal{U}^*}([0, 1])} \geq 1 - \frac{2K\epsilon'}{K\tau_K} = 1 - \frac{2\epsilon'}{\tau_K}.$$

Thus, if we set $\epsilon' \leq \frac{\tau_K}{2}\epsilon$ when calling the $\mathsf{FullCATE}$ problem, we obtain a $(1-\epsilon, \delta)$-OPT allocation function. Hence, we have solved the $\mathsf{ALLOC}$ problem, as required. $\qquad\square$

## B  DEFERRED PROOFS FROM SECTION 3

**Claim B.1.** *Given a budget $K$, $\mathcal{U}^* = \{u \mid \tau(u) \geq \tau_K\}$.*

*Proof.* By definition of $\tau_K$, there are exactly $K - 1$ units that have $\tau(u)$ value higher than $\tau_K$, all of which are included in $\mathcal{U}$. Hence swapping any unit in $\mathcal{U}^*$ with one not in the $\mathcal{U}^*$ would decrease the value of the allocation, which would contradict the optimality of $\mathsf{ALLOC}^*$. $\qquad\square$

**Lemma 4.** *Given any $\mathcal{X}, \mathcal{U}, K$, Algorithm 1 requires $O(M \ln(2M/\delta)/\epsilon)$ many samples from $\mathcal{X}$.*

*Proof.* By a Hoeffding bound using parameters $\rho$ and $\delta/M$, letting $N(u)$ denote the number of samples that we require from unit $u$, we have that

$$\Pr\left[\left|\hat{\tau}(u) - \tau(u)\right| \geq \rho\right] \leq 2\exp(-2N(u) \cdot \rho^2) = \frac{\delta}{M} \implies N(u) = \frac{\ln(2M/\delta)}{2\rho^2}.$$

Therefore, adding the number of samples across the $M$ units, we obtain that this low-accuracy estimation requires

$$\frac{M \ln(2M/\delta)}{\rho^2} = \frac{M \ln(2M/\delta)}{\epsilon}$$

many samples from $\mathcal{X}$. $\qquad\square$

**Lemma 5.** *The output estimates $\hat{\tau}_K$ and the set $\mathcal{U}_{\mathsf{LEA}}$ returned by Algorithm 1 satisfy the following properties: (1) $|\hat{\tau}_K - \tau_K| \leq \rho$. (2) All units $u$ such that $\tau(u) > \tau_K + 2\rho$ belong to $\mathcal{U}_{\mathsf{LEA}}$. (3) All units $u$ such that $\tau(u) < \tau_K - 2\rho$ do not belong to $\mathcal{U}_{\mathsf{LEA}}$.*

*Proof.* We prove each of the three parts separately.

(1) By definition of $\tau_K$, there are exactly $K$ values $\tau(u)$ such that $\tau(u) \geq \tau_K$. By the $\rho$-accuracy guarantee on our estimates $\hat{\tau}(u)$, it follows that $\hat{\tau}(u) \geq \tau_K - \rho$ for all such $u$. Then, there are $K$ values $\hat{\tau}(u)$ that are at least $\tau_K - \rho$. By definition of $\hat{\tau}_K$, then, it must be that $\hat{\tau}_K \geq \tau_K - \rho$.

Symmetrically, $\hat{\tau}_K$ is definitionally the $(N - K + 1)$-th smallest number in the increasing sequence of values $\hat{\tau}(u)$, and similarly $\tau_K$ is the $(N - K + 1)$-th smallest number in the increasing sequence of values $\tau(u)$. Therefore, exactly $N - K$ values $\tau(u)$ are such that $\tau(u) < \tau_K$. By the $\rho$-accuracy guarantee, $\hat{\tau}(u) < \tau_K + \rho$ for all such $u$. Thus there are $N - K$ values $\hat{\tau}(u)$ that are below $\tau_K + \rho$. By definition of $\hat{\tau}_K$, it must be that $\hat{\tau}_K \leq \tau_K + \rho$.

Since $\hat{\tau}_K \geq \tau_K - \rho$ and $\hat{\tau}_K \leq \tau_K + \rho$, it follows that $\hat{\tau}_K \in [\tau_K - \rho, \tau_K + \rho]$. Hence $|\hat{\tau}_K - \tau_K| \leq \rho$.

(2) By the $\rho$-accuracy guarantee, all units $u$ satisfying $\tau(u) > \tau_K + 2\rho$ also satisfy $\hat{\tau}(u) > \tau_K + \rho$. By part (1) of this claim, it then follows that $\hat{\tau}(u) > \hat{\tau}_K$. Hence, Algorithm 1 selects such $u$ for treatment and so they belong to $\mathcal{U}_{\mathsf{LEA}}$.

(3) Symmetrically, by the $\rho$-accuracy guarantee, all units $u$ satisfying $\tau(u) < \tau_K - 2\rho$ also satisfy $\hat{\tau}(u) < \tau_K - \rho$. By part (1) of this claim, it then follows that $\hat{\tau}(u) < \hat{\tau}_K$. Hence, Algorithm 1 does not select such $u$ for treatment and so they do not belong to $\mathcal{U}_{\mathsf{LEA}}$. $\qquad\square$

**Claim 3.1.** $\dfrac{V_{\mathcal{U}_{\mathsf{LEA}}}([0,1])}{V_{\mathcal{U}^*}([0,1])} \geq 1 - \dfrac{4\rho K_0}{V(A_1) + (\tau_K + 2\rho)K_0}.$

Before we prove Claim 3.1, we show some intermediate lemmas.

First, we use interval $A_1$ to analyze the difference between $V_{\mathcal{U}_{\mathsf{LEA}}}$ and $V_{\mathcal{U}^*}$.

**Claim B.2.** $V_{\mathcal{U}^*}([0,1]) = V_{\mathcal{U}^*}([\tau_K, 1]).$

*Proof.* As shown in Claim B.1, the optimal allocation for budget $K$ contains all units $u$ such that $\tau(u) \geq \tau_K$. $\qquad\square$

**Claim B.3.** $V_{\mathcal{U}_{\mathsf{LEA}}}([0,1]) \geq V_{\mathcal{U}_{\mathsf{LEA}}}(A_1) = V((\tau_K + 2\rho, 1]).$

*Proof.* Firstly, $V_{\mathcal{U}_{\mathsf{LEA}}}([0,1]) \geq V_{\mathcal{U}_{\mathsf{LEA}}}(A_1)$ follows since $A_1 \subseteq [0,1]$ and $V(\cdot)$ is an additive function. Secondly, $V_{\mathcal{U}_{\mathsf{LEA}}}(A_1) = V((\tau_K + 2\rho, 1])$ follows by the definition of $A_1$. $\qquad\square$

We can lower bound $V_{\mathcal{U}_{\mathsf{LEA}}}([0,1])$ more precisely using the interval $D = [\tau_K, \tau_K + 2\rho]$ and the set $\mathcal{U}_{\mathsf{LEA}}^0$ (see Definition 7). Specifically, we write the values of the optimal allocation and our allocation using the $D$ interval.

**Claim B.4.** *(1)* $V_{\mathcal{U}^*}([0,1]) = V_{\mathcal{U}^*}(D) + V_{\mathcal{U}^*}((\tau_K + 2\rho, 1]).$

*(2)* $V_{\mathcal{U}_{\mathsf{LEA}}}([0,1]) = \sum_{i \in \mathcal{U}_{\mathsf{LEA}}^0} \tau(i) + V_{\mathcal{U}_{\mathsf{LEA}}}((\tau_K + 2\rho, 1]).$

*Proof.* The first equality follows by Claim B.2 and the additivity of $V(\cdot)$. The second equality follows just from the additivity of $V(\cdot)$. $\qquad\square$

We want to separate the budget between the number of units it treats over $A_1$ versus elsewhere, so we split $K$ as follows:

**Definition 11.** *Let* $K_1 = |\{u \mid \tau(u) \in A_1\}|$, *and let* $K_0 = K - K_1$.

We can now provide the following bounds:

**Claim B.5.** $V_{\mathcal{U}_{\mathsf{LEA}}^0}([0,1]) \geq (\tau_K - 2\rho)K_0.$

*Proof.* By Lemma 5, all units $u \in \mathcal{U}_{\mathsf{LEA}}$ satisfy $\tau(u) \geq \tau_K - 2\rho$, and $K_0 \leq K$. $\qquad\square$

**Claim B.6.** $V_{\mathcal{U}^*}(D) \leq (\tau_K + 2\rho)K_0$.

*Proof.* By definition of $A_1$, all units $u$ such that $\tau(u) \in A_1$ satisfy satisfy $\tau(u) > \tau_K + 2\rho$. Hence, by definition of $K_0$, any unit $u \in K_0$ must satisfy $\tau(u) \leq \tau_K + 2\rho$. $\qquad\square$

**Claim B.7.** $\dfrac{V_{\mathcal{U}_{\mathsf{LEA}}}\big([0,1]\big)}{V_{\mathcal{U}^*}\big([0,1]\big)} = \dfrac{V_{\mathcal{U}_{\mathsf{LEA}}}(A_1) + V_{\mathcal{U}_{\mathsf{LEA}}^0}\big([0,1]\big)}{V_{\mathcal{U}^*}(A_1) + V_{\mathcal{U}^*}(D)} \geq \dfrac{V_{\mathcal{U}_{\mathsf{LEA}}}(A_1) + (\tau_K - 2\rho)K_0}{V_{\mathcal{U}^*}(A_1) + (\tau_K + 2\rho)K_0}.$

*Proof.* The first equality follows from Claim B.4; the inequality from Claims B.5, and B.6. $\qquad\square$

To further simplify the expression, we use:

**Claim B.8.** $V_{\mathcal{U}_{\mathsf{LEA}}}(A_1) = V_{\mathcal{U}^*}(A_1)$.

*Proof.* By definition of $A_1$, all units $u \in A_1$ satisfy $\tau(u) > \tau_K + 2\rho$. By definition of $\tau_K$, all such $u$ are part of $\mathcal{U}^*$. As for our allocation, by Lemma 5 all such $u$ are also part of $\mathcal{U}_{\mathsf{LEA}}$. $\qquad\square$

By the previous claim, we can use $V(A_1)$ as a short-hand for $V_{\mathcal{U}_{\mathsf{LEA}}}(A_1) = V_{\mathcal{U}^*}(A_1)$. With these intermediate small claims, we can now show Claim 3.1.

*Proof of Claim 3.1.* This follows from re-arranging the expression in Claim B.7. Specifically,

$$
\begin{aligned}
\frac{V_{\mathcal{U}_{\mathsf{LEA}}}(A_1) + (\tau_K - 2\rho)K_0}{V_{\mathcal{U}^*}(A_1) + (\tau_K + 2\rho)K_0} &= \frac{V(A_1) + (\tau_K - 2\rho)K_0}{V(A_1) + (\tau_K + 2\rho)K_0} \\
&= \frac{V(A_1) + (\tau_K + 2\rho)K_0 - (\tau_K + 2\rho)K_0 + (\tau_K - 2\rho)K_0}{V(A_1) + (\tau_K + 2\rho)K_0} \\
&= 1 - \frac{(\tau_K + 2\rho)K_0 - (\tau_K - 2\rho)K_0}{V(A_1) + (\tau_K + 2\rho)K_0} \\
&= 1 - \frac{4\rho K_0}{V(A_1) + (\tau_K + 2\rho)K_0}
\end{aligned}
$$

$\qquad\square$

For the proof of Claim 3.2, recall the formal definitions of $\theta_K$ and $\gamma_1$:

**Definition 12.** *Let* $\Pr_\tau[\tau_K, \tau_K + 2\rho] = \theta_K$ *and let* $V(A_1) = \gamma_1 M$.

Moreover, we can relate $K_0$ to the distribution $\Pr_\tau$ of treatment effect values:

**Claim B.9.** $K_0 = \Pr_\tau\big[[\tau_K, \tau_K + 2\rho]\big] \cdot M$.

*Proof.* All units $u$ selected with the $K_0$ budget satisfy $\tau(u) \notin A_1$ by definition, and hence they satisfy $\tau(u) \leq \tau_K + 2\rho$. Since all units selected with the $K_0$ budget are still part of $\mathcal{U}^*$, it follows that they all satisfy $\tau(u) \geq \tau_K$. Hence, $|K_0| = \big|\{u \mid \tau(i) \in [\tau_K, \tau_K + 2\rho]\big|$. Lastly, $\big|\{u \mid \tau(i) \in [\tau_K, \tau_K + 2\rho]\big| = \Pr_\tau[\tau_K, \tau_K + 2\rho] \cdot N$ by definition of $\Pr_\tau$. $\qquad\square$

**Claim 3.2.** *Given any* $\mathcal{X}, \mathcal{U}, K$, *Algorithm 1 returns a* $\left(1 - \dfrac{4\gamma\theta_K}{\gamma_1 + \tau_K\theta_K} \cdot \sqrt{\epsilon}\right)$*-optimal approximation with* $O(M\ln(2M/\delta)/\epsilon)$ *many samples.*

*Proof.* Using the Definition 12 on Claim 3.1 and dividing by $N$ it follows that

$$
\frac{V_{\mathcal{U}_{\mathsf{LEA}}}\big([0,1]\big)}{V_{\mathcal{U}^*}\big([0,1]\big)} \geq 1 - \frac{4\rho\theta_K}{\gamma_1 + (\tau_K + 2\rho)\theta_K}.
$$

This is a $(1 - \epsilon)$-OPT approximation if

$$\frac{4\rho\theta_K}{\gamma_1 + (\tau_K + 2\rho)\theta_K} \leq \epsilon.$$

Hence, to get a $(1 - \epsilon)$-OPT approximation, we require

$$\theta_K \leq \frac{\epsilon\gamma_1}{4\rho - \epsilon(\tau_K + 2\rho)}.$$

Alternatively, by plugging in $\rho = \gamma\sqrt{\epsilon}$, we obtain the expression in the statement of Claim 3.2. $\square$

**Claim 3.3.** *For any budget $K \leq M$, we can compute $\tau_K, \gamma_1, V_{\mathcal{U}^*}$ as: (1) $\tau_K = F^{-1}(1 - K/M)$, (2) $\gamma_1 = \frac{1}{M} \int_{\tau_K + 2\rho}^1 t \, dF(t)$, (3) $V_{\mathcal{U}^*}([0, 1]) = \int_{\tau_K}^1 t \, dF(t)$.*

*Proof.* First, $\tau_K$ is definitionally the $K$-th largest $\tau(u)$ value. In terms of the CDF $F$ of treatment effect values $\Pr_\tau$, this means that $\tau_K$ corresponds to the value in $[0, 1]$ satisfying $F(\tau_K) = 1 - K/M$.

Second, $\gamma_1$ definitionally satisfies $\gamma_1 = V(A_1)/N$. By the definition of the value function and given that $A_1 = (\tau_K + 2\rho, 1]$, it follows that

$$V(A_1) = \sum_{u | \tau(u) \in A_1} \tau(u).$$

Therefore, this corresponds to taking the integral $\int_{\tau_K + 2\rho}^1 t f(t) \, dt$, where $t$ accounts for the value of $\tau(u)$ and $f(t)$ for the density of units for each value.

Similarly,

$$V_{\mathcal{U}^*}([0, 1]) = \int_0^1 t f(t) \, dt = \int_{\tau_K}^1 t f(t) \, dt,$$

given that by Claim B.2 $V_{\mathcal{U}^*}([0, 1]) = V_{\mathcal{U}^*}([\tau_K, 1])$. Lastly, integrating $t f(t) \, dt$ with the PDF is equivalent to integrating $t \, dF(t)$ with the CDF. $\square$

**Claim 3.4.** *For any $t \in [0, 1]$, $\hat{F}(t - \rho) \leq F(t) \leq \hat{F}(t + \rho)$.*

*Proof.* If $\hat{\tau}(u) \leq t - \rho$, by the $\rho$-accuracy guarantee it follows that $\tau(u) \leq \hat{\tau}(u) + \rho \leq t$. Hence:

$$\hat{F}(t - \rho) = \frac{|\{u : \hat{\tau}(u) \leq t - \rho\}|}{M} \leq \frac{|\{u : \tau(u) \leq t\}|}{M} = F(t).$$

Symmetrically, if $\tau(u) \leq t$, then by the $\rho$-accuracy guarantee it follows that $\hat{\tau}(u) \leq \tau(u) + \rho \leq t + \rho$. Hence:

$$F(t) = \frac{|\{u : \tau(u) \leq t\}|}{M} \leq \frac{|\{u : \hat{\tau}(u) \leq t + \rho\}|}{M} = \hat{F}(t + \rho).$$

$\square$

We can equivalently express Claim 3.4 in terms of the number of units present in an interval. Specifically, for any interval in $[0, 1]$ we have that:

**Claim B.10.** *Given any interval $[a, b] \subseteq [0, 1]$,*

$$\left| u : \hat{\tau}(u) \in [a + \rho, b - \rho] \right| \leq \Pr_\tau \left[ [a, b] \right] \cdot M \leq \left| u : \hat{\tau}(u) \in [a - \rho, b + \rho] \right|.$$

*Proof.* By the $\rho$-accuracy guarantee, we know that for any unit $u$, if $\hat{\tau}(u) \in [a + \rho, b - \rho]$, then the corresponding $\tau(u)$ is in $[a, b]$. Similarly, if $\tau(u) \in [a, b]$, then this implies that $\hat{\tau}(u) \in [a - \rho, b + \rho]$. Lastly, the number of $\tau(u)$ such that $\tau(u) \in [a, b]$ is precisely equal to $\Pr_\tau \left[ [a, b] \right] \cdot M$. $\square$

This allows us to use our low-accuracy estimates of the treatment effects to approximate the true probability mass in any interval by enlarging or shrinking the interval accordingly.

We can use this approximation of the CDF to check the smoothness behavior of the treatment effect values from our low-accuracy estimates $\hat{\tau}_K$. Specifically, we are interested in the number of units in the $D = [\tau_K, \tau_K + 2\rho]$ interval. By Claim B.10, we have that

$$\Pr_{\tau}\left[[\tau_K, \tau_K + 2\rho]\right] \cdot M \leq \left|u : \hat{\tau}(u) \in [\tau_K - \rho, \tau_K + 3\rho]\right|.$$

Moreover, by the $\rho$-accuracy guarantee, we know that $\tau_K \in [\hat{\tau}_K - \rho, \hat{\tau}_K + \rho]$. Hence, the number of units that have $\hat{\tau}(u)$ value in the interval $[\hat{\tau}_K - 2\rho, \hat{\tau}_K + 4\rho]$ upper bounds the quantity

$$\Pr_{\tau}\left[[\tau_K, \tau_K + 2\rho]\right] \cdot M,$$

and can be entirely computed from our low-accuracy estimates $\hat{\tau}$. This allows us to check whether the interval around the optimal threshold contains too many units, and thus fails to satisfy the smoothness condition required by $\rho$-regularity. As we discuss in Section 5, we can also use our low-accuracy estimation of the CDF to guide the choice of the budget, provided there is flexibility in the choice of the budget.

## C    DEFERRED PROOFS FROM SECTION 4

We begin by showing our upper-bound for $\rho$-regular distributions, and then show the upper-bound for uniform distributions as a corollary, given that the uniform distribution is $\rho$-regular.

**Theorem 7.** *Let the treatment effect values be distributed according to a $\rho$-regular distribution for $\rho = \Theta(\sqrt{\epsilon})$. Then, we can obtain a $(1 - \epsilon, \delta)$-approximation of the optimal allocation with $N = O(M \ln(2M/\delta)/\epsilon)$ many samples from $\mathcal{X}$ for any budget $0 \leq K \leq M$.*

*Proof.* For the first part, we use our Algorithm 1. The sample complexity follows from Claim 4. As for the accuracy guarantee, by Claim 3.1 we know that we get a $(1 - \epsilon, \delta)$-OPT approximation if

$$\frac{4\rho\theta_K}{\gamma_1 + (\tau_K + 2\rho)\theta_K} \leq \epsilon. \tag{2}$$

Because $\Pr_{\tau}$ is $\rho$-regular, we have that

$$\theta_K = \Pr_{\tau}[\tau_K, \tau_K + 2\rho] \leq 2\rho c$$

for some $c = \Theta(1)$. As in Section 4, we bound the value of $V(A_1)$. We obtain a more conservative bound by dropping $(\tau_K + 2\rho)\theta_K$ from the denominator and requiring that

$$\frac{4\rho(2\rho c)}{\gamma_1} = \frac{8c\rho^2}{\gamma_1} \leq \epsilon.$$

Hence, we obtain a $(1 - \epsilon)$-OPT approximation by setting $\gamma = \sqrt{\gamma_1/(8c)}$ and $\rho = \gamma\sqrt{\epsilon}$, which ensures that the previous inequality holds.

We can also keep the more precise ratio $\frac{4\rho\theta_K}{\gamma_1 + (\tau_K + 2\rho)\theta_K}$. Note that

$$\gamma_1 + (\tau_K + 2\rho)\theta_K \geq V_{\mathcal{U}^*}([\tau_K, 1]) = V_{\mathcal{U}^*}.$$

Note that $V_{\mathcal{U}^*}$ is a function of $K$ as well (even if not indicated with a subscript). Therefore, we need to satisfy the inequality

$$\frac{4\rho\theta_K}{\gamma_1 + (\tau_K + 2\rho)\theta_K} \leq \frac{4\rho\theta_K}{V_{\mathcal{U}^*}} \leq \epsilon,$$

which we do by setting $\gamma = \sqrt{V_{\mathcal{U}^*}/(8c)}$ and $\rho = \gamma\sqrt{\epsilon}$.

$\square$

It is clear from this proof why we need $\theta_K \leq 2\rho c$. If we only have the bound $\theta_K \leq 1$, then we get $4\rho/V_{\mathcal{U}^*} \leq \epsilon$, which in turn requires $\rho$ to be of the order of $\epsilon$, rather than of $\sqrt{\epsilon}$. But if $\theta_K \leq 2\rho c$, then we get the term $\rho^2$ in the numerator, which is the crucial point that allows us to get $\rho = \Theta(\sqrt{\epsilon})$ and thus an optimal allocation algorithm with sample complexity $O(M/\epsilon)$ rather than with $O(M/\epsilon^2)$.

**Theorem 6.** *Let the treatment effect values be uniformly distributed. Then: (1) We can obtain a $(1 - \epsilon, \delta)$-approximation of the optimal allocation with $N = O(M \ln(2M/\delta)/\epsilon)$ many samples from $\mathcal{X}$ for any budget $K \leq M$. (2)* FullCATE *requires $\Omega(M/\epsilon^2)$ samples.*

*Proof.* For the first part, we use our Algorithm 1. The sample complexity follows from Claim 4. As for the accuracy guarantee, by Claim 3.1 we know that we get a $(1 - \epsilon, \delta)$-OPT approximation if

$$\frac{4\rho\theta_K}{\gamma_1 + (\tau_K + 2\rho)\theta_K} \leq \epsilon. \tag{3}$$

For a fixed $M$, recall that for a uniform distribution of $M$ treatment effect values, the $\tau(u)$ are equally-spaced in $[0, 1]$ and randomly permuted among them. Thus, this discrete version of the uniform distribution is $\rho$-regular with $\rho = 1/M$ and $c = 2$. This follows from the following argument: let $S \subseteq [0, 1]$ be any interval of length $\ell \geq 2\rho$. By the discrete equal spacing of the treatment effects, there are at most $\ell M + 2$ units in $S$, and so $\Pr[S] \leq (\ell M + 2)/M = \ell + 2/M$ under this distribution. So for all intervals $S$ with $|S| \geq 2\rho = 2/M$, it follows that

$$\Pr[S] \leq \ell + 2/M \leq 2\ell = 2\mathcal{U}[S],$$

thus satisfying the definition of $\rho$-regularity with $c = 2$. Given that $M$ is fixed, $\rho = \Theta(\sqrt{\epsilon})$ and thus we can apply Theorem 7 to this discrete uniform distribution, from which the upper bound follows.

For the lower bound, we provide a minimax argument based on LeCam's method. Given a parameter space $\Theta$, for each $\theta \in \Theta$ we let $P_\theta$ denote the distribution of the observed data under parameter $\theta$. Then, LeCam's Theorem (also known as the *two-point method*) states that

$$\inf_\Psi \sup_{\theta, \theta'} \max\{\Pr_{P_\theta}[\Psi \neq 0], \Pr_{P_{\theta'}}[\Psi \neq 1] \geq \frac{1}{2} e^{-\text{KL}(P_\theta || P_{\theta'})}$$

for any estimator $\Psi$ and parameter values $\theta, \theta' \in \Theta$, where $\Pr_{P_\theta}$ denotes the probability when the data is distributed according to $P_\theta$, and similarly for $P_{\theta'}$. Here, $\text{KL}(P_\theta || P_{\theta'})$ denotes the KL divergence between $P_\theta$ and $P_{\theta'}$.

Because we are working with the discrete uniform distribution, where the $\tau(u)$ values are uniformly spaced in $[0, 1]$, we consider the family of values $p = \{1/M, 2/M, \ldots, 1\}$. Let $p$ be one of such values, bounded away from 0 and from 1; without loss of generality, we can let $p \in [1/4, 3/4]$. For a unit $u$, we consider the two parameters

$$\theta : \tau(u) = p - \epsilon, \quad \theta' : \tau(u) = p + \epsilon.$$

If $\hat{\tau}(u)$ is an $(\epsilon, \delta)$-accurate estimate of $\tau(u)$, then definitionally it must satisfy

$$\Pr\left[|\hat{\tau}(u) - (p - \epsilon)| \leq \epsilon\right] \geq 1 - \delta, \quad \Pr\left[|\hat{\tau}(u) - (p + \epsilon)| \leq \epsilon\right] \geq 1 - \delta.$$

We consider the test $\Psi_u = \mathbf{1}[\hat{\tau}(u) \geq p]$. For an $(\epsilon, \delta)$-accurate estimate, we have that

$$\Pr_\theta[\Psi_u = 1] \leq \delta, \quad \Pr_{\theta'}[\Psi_u = 0] \leq \delta.$$

That is, a good estimator yields a test with small error.

For each unit $u$, we observe $N$ samples drawn i.i.d. from $\text{Bern}(\tau(u))$. By standard bounds on the KL divergence we have that

$$\text{KL}(\text{Bern}(p - \epsilon) || \text{Bern}(p + \epsilon)) \leq 13\epsilon^2.$$

Therefore, LeCam's Theorem tells us that for any estimator $\Psi$,

$$\Pr_\theta[\Psi = 1] + \Pr_{\theta'}[\Psi = 0] \geq \frac{1}{2} e^{-\text{KL}(\text{Bern}(p-\epsilon)||\text{Bern}(p+\epsilon))}.$$

Hence we get that

$$2\delta \geq \frac{1}{2} e^{-13N\epsilon^2} \implies N \geq \frac{1}{13\epsilon^2} \log(1/4\delta).$$

Lastly, because the $\tau(u)$ values are uniformly distributed over $[0, 1]$ it follows that $\Theta(M)$ many units have $\tau(u)$ value bounded away from 0 and 1. (In the case where we consider $p \in [1/4, 3/4]$, there are $M/2$ many such units.) Therefore, by applying the LeCam Theorem on each such unit and taking a union bound over them it follows that the total number of samples required for producing $(\epsilon, \delta)$-accurate estimates for all units that have $\tau(u)$ value bounded away from 0 and 1 is $N = O((M/\epsilon^2) \log(M/\delta))$ for any estimator $\Psi$, and hence $N = \Omega(M/\epsilon^2)$, as we wanted to show. $\qquad\square$

While the upper bound of Theorem 6 follows as a corollary of the $\rho$-regularity theorem (Theorem 7) for the discrete uniformly-spaced placement of the treatment effect values, it is illustrative to prove the upper bound for an idealized version of the uniform distribution, where $M \to 0$.

In this case, we have that

$$\theta_K = \Pr_\tau[\tau_K, \tau_K + 2\rho] = 2\rho.$$

Recall that $\gamma_1$ is such that $V(A_1) = \gamma_1 N$. We can compute $V(A_1)$ exactly (as per Claim 3.3), but for this proof it is enough to bound $V(A_1)$. Given that $A_1 = (\tau_K + 2\rho, 1]$, it follows that $\Pr_\tau[A_1] = 1 - (\tau_K + 2\rho) = 1 - \tau_K - 2\rho$. Hence:

$$\gamma_1 = \frac{V(A_1)}{N} \geq (\tau_K + 2\rho) \Pr_\tau[A_1] = (\tau_K + 2\rho)(1 - \tau_K - 2\rho),$$

given that all units in $A_1$ have value at least $\tau_K + 2\rho$. Using this bound on the denominator of the LHS in Equation 3 we get that

$$\gamma_1 + (\tau_K + 2\rho)\theta_K \geq (\tau_K + 2\rho)(1 - \tau_K - 2\rho) + (\tau_K + 2\rho)(2\rho) = (\tau_K + 2\rho)(1 - \tau_K).$$

Hence by Equation 3 we obtain a $(1 - \epsilon, \delta)$-OPT approximation if

$$\frac{4\rho\theta_K}{\gamma_1 + (\tau_K + 2\rho)\theta_K} = \frac{4\rho(2\rho)}{(\tau_K + 2\rho)(1 - \tau_K)} = \frac{8\rho^2}{(\tau_K + 2\rho)(1 - \tau_K)} \leq \epsilon.$$

This is true whenever $\rho \leq \sqrt{\frac{\tau_K(1 - \tau_K)}{8}}\sqrt{\epsilon}$. In the case of the uniform distribution, we have that $\tau_K = 1 - K/M$, which allows us to compute the value of $\gamma$ exactly. We can also provide a general bound for all budgets $K$: the function $\tau_K(1 - \tau_K)$ for $\tau_K \in [0, 1]$ is maximized at $1/4$, and so setting $\rho = \gamma\sqrt{\epsilon}$ for $\gamma = \frac{1}{4\sqrt{2}}$ guarantees that our Algorithm 1 achieves a $(1 - \epsilon, \delta)$-OPT approximation.

**Theorem 8.** *Let $c$ be the minimum value such that $\Pr_\tau[S] \leq c \cdot \mathcal{U}[S]$ for all subsets $S \subseteq [0, 1]$ such that $|S| \geq 2\rho$. For any budget $K \leq M$, we call Algorithm 1 with with $\rho = \sqrt{V_{\mathcal{U}^*}/8c}$. This gives a $(1 - \epsilon)$-OPT approximation of the optimal allocation with $O(M \ln(2M/\delta)/\rho^2)$ many samples.*

*Proof.* While $c$ might not be of the order of $\Theta(1)$, we can still write $\theta_K = 2\rho c$. Then, we get a $(1 - \epsilon, \delta)$-OPT approximation if

$$\frac{4\rho\theta_K}{\gamma_1 + (\tau_K + 2\rho)\theta_K} = \frac{8c\rho^2}{\gamma_1 + (\tau_K + 2\rho)(2\rho c)} \leq \epsilon.$$

We can similarly ensure the more conservative inequality $8c\rho^2/\gamma_1 \leq \epsilon$ by setting $\gamma = \sqrt{\frac{\gamma_1}{8c}}$ and $\rho = \gamma\sqrt{\epsilon}$. Less conservatively, we can set $\gamma = \sqrt{V_{\mathcal{U}^*}^K/(8c)}$, given that $\gamma_1 + (\tau_K + 2\rho)\theta_K \geq V_{\mathcal{U}^*}$. $\square$

## C.1 EXAMPLES OF $\gamma$ FOR TYPICAL DISTRIBUTIONS

For a distribution $\Pr_\tau$, recall that we denote the CDF by $F$, and so we denote the density function by $f$. Note that we can bound the value of $\gamma_1$ using $f$:

$$\begin{aligned}
\gamma_1 = \int_{\tau_K + 2\rho}^1 t f(t)\, dt &= \int_{\tau_K}^1 t f(t)\, dt - \int_{\tau_K}^{\tau_K + 2\rho} t f(t)\, dt \\
&\geq \int_{\tau_K}^1 t f(t)\, dt - (\tau_K + 2\rho)\theta_K \\
&\geq \int_{\tau_K}^1 t f(t)\, dt - (\tau_K + 2\rho)2c\rho.
\end{aligned}$$

Recall that $\int_{\tau_K}^1 t f(t)\, dt = V_{\mathcal{U}^*}^K$. Indeed, in order to compute the value of the optimal allocation, it suffices to have the CDF function, with any permutation of the units. Similarly, we can compute $\tau_K$ directly from the CDF. Following our expressions derived in Claim 3.3, we can compute the values of $\gamma, c$, and $V_{\mathcal{U}^*}$ for various typical distributions from their CDFs. In these examples, it is neater to integrate $t f(t)$ instead of the CDF directly, but not that we only need knowledge of the CDF of the distribution (and the integration is equivalent).

**Uniform distribution.** For the uniform distribution, we have $f(t) = 1$. Hence, $c = 1$, since it is the maximum possible value of $f(t)$. Moreover, we have:

$$V_{\mathcal{U}^*}([0,1]) = \int_{\tau_K}^1 t f(t)\, dt = \int_{\tau_K}^1 t\, dt = \frac{1 - \tau_K^2}{2}.$$

Moreover, $\tau_K = 1 - K/M$. Therefore, by Theorem 7 and Claim 8, it follows that

$$\gamma = \sqrt{\frac{V_{\mathcal{U}^*}}{8c}} = \sqrt{\frac{1 - (1 - K/M)^2}{8}}.$$

For all $K \in [0, N]$, $\gamma$ is maximized at $\approx 0.35$.

**Beta distribution.** For $\alpha, \beta$, the PDF of the Beta distribution is given by

$$f(t; \alpha, \beta) = \frac{1}{B(\alpha, \beta)} t^{\alpha-1}(1-t)^{\beta-1},$$

where the Beta function $B(\alpha, \beta)$ is the normalization constant, and the CDF is given by

$$F(t) = \frac{B(t; \alpha, \beta))}{B(\alpha, \beta)} := I_x(\alpha, \beta),$$

where $B(x; \alpha, \beta) = \int_0^x t^{\alpha-1}(1-t)^{\beta-1}\, dt$ is known as the *incomplete Beta function*.

For $\alpha, \beta > 1$, the mode of the distribution is given by $\frac{\alpha-1}{\alpha+\beta-2}$ (if $\alpha = \beta$, then the mode is equal to $1/2$), and so

$$c \le \sup_t f(t) = f\left(\frac{\alpha-1}{\alpha+\beta-2}\right) = \frac{(\alpha-1)^{\alpha-1}(\beta-1)^{\beta-1}}{(\alpha+\beta-2)^{\alpha+\beta-2}B(\alpha, \beta)}.$$

As for $V_{\mathcal{U}^*}$, we have that

$$\begin{aligned}
V_{\mathcal{U}^*}([0,1]) = \int_{\tau_K}^1 t f(t)\, dt &= \int_{\tau_K}^1 t \cdot \frac{1}{B(\alpha, \beta)} t^{\alpha-1}(1-t)^{\beta-1} \\
&= \frac{1}{B(\alpha, \beta)} \int_{\tau_K}^1 t^\alpha (1-t)^{\beta-1}\, dt \\
&= \frac{B(\alpha+1, \beta) - B(\tau_K; \alpha+1, \beta)}{B(\alpha, \beta)} \\
&= \frac{B(\alpha+1, \beta)}{B(\alpha, \beta)}\left(1 - I_{\tau_K}(\alpha+1, \beta)\right) \\
&= \frac{a}{a+b}\left(1 - I_{\tau_K}(\alpha+1, \beta)\right).
\end{aligned}$$

Lastly, $\tau_K$ is the value such that $F(\tau_K) = 1 - K/M$, or equivalently, such that $\int_{\tau_K}^1 f(t)\, dt = 1 - K/M$. Given that $F(t) = I_t(\alpha, \beta)$, it follows that

$$\tau_K = I_{1-\alpha}^{-1}(\alpha, \beta).$$

For example, for Beta$(2, 2)$ we have that $f(t) = 6t(1 - t)$ and so $\sup_t f(t) = f(1/2) = 1.5$. We also have that $I_t(3, 2) = 4t^3 - 3t^4$, and so

$$V_{\mathcal{U}^*} = \frac{2}{4}\left(1 - I_{\tau_K}(\alpha+1, \beta)\right) = \frac{2}{4}\left(1 - (4\tau_K^3 - 3\tau_K^4)\right) = \frac{1}{2}(1 - 4\tau_K^3 + 3\tau_K^4).$$

Given that $F(t) = I_t(2, 2) = 3t^2 - 2t^3$, and that $\tau_K$ is the value in $[0, 1]$ such that $F(\tau_K) = 1 - K/M$, it follows that $\tau_K$ corresponds to the unique root of the polynomial $2\tau_K^3 - 3\tau_K^2 + 1 - K/M$ in $[0, 1]$. For example, if $K/M = 0.5$, we get that $V_{\mathcal{U}^*} \approx 0.34$ and $\gamma \le 0.17$. For $K/M = 0.25$ we get $\gamma \le 0.12$, and for $K/M = 0.75$ we get $\gamma \le 0.19$.

For Beta$(3, 3)$ we have that $f(t) = 30t^2(1 - t)^2$ and so $\sup_t f(t) = f(1/2) = 1.875$. We also have that $I_t(4, 3) = 10t^4 - 15t^5 + 6t^6$, and so

$$V_{\mathcal{U}^*} = \frac{3}{6}\left(1 - I_{\tau_K}(\alpha+1, \beta)\right) = \frac{1}{2}\left(1 - (10\tau_K^4 - 15\tau_K^5 + 6\tau_K^6)\right) = \frac{1}{2}(1 - 10\tau_K^4 + 15\tau_K^5 - 6\tau_K^6).$$

Given that $F(t) = I_t(3,3) = 10t^3 - 15t^4 + 6t^5$ and $t_K$ satisfies $F(t_K) = 1 - K/M$, it follows that $\tau_K$ corresponds to the root of the polynomial $6\tau_K^5 - 15\tau_K^4 + 10\tau_K^3 - (1 - K/M) = 0$ in $[0,1]$. For example, if $K/M = 0.5$, we get that $V_{\mathcal{U}^*} \approx 0.37$ and $\gamma \leq 0.16$. For $K/M = 0.25$ we get $\gamma \leq 0.13$, and for $K/M = 0.75$ we get $\gamma \leq 0.17$.

For Beta$(2,4)$ we have that $f(t) = 20t(1-t)^3$ and mode $\frac{2-1}{2+4-2} = \frac{1}{4}$, and so $c \leq \sup_t f(t) = f(1/4) \approx 2.1$. We also have that $I_t(3,4) = 20t^3 - 45t^4 + 36t^5 - 10t^6$, and so

$$V_{\mathcal{U}^*} = \frac{2}{6}\big(1 - I_{\tau_K}(\alpha+1, \beta)\big) = \frac{1}{3}\big(1 - (20t^3 - 45t^4 + 36t^5 - 10t^6)\big) = \frac{1}{3}\big(1 - 20t^3 + 45t^5 - 36t^6 + 10t^6\big).$$

Using $F(t) = I_t(2,4)$, it follows that $\tau_K$ corresponds to the root of the polynomial $10\tau_K^2 - 20\tau_K^3 + 15\tau_K^4 - 4\tau_K^5 - 1 + K/M$ in $[0,1]$. For example, if $K/M = 0.5$, we get that $V_{\mathcal{U}^*} \approx 0.24$ and $\gamma \leq 0.12$. For $K/M = 0.25$ we get $\gamma \leq 0.09$, and for $K/M = 0.75$ we get $\gamma \leq 0.13$.

**Gaussian distribution.** Given a normal distribution $\mathcal{N}(\mu, \sigma^2)$, we truncate it to $[0,1]$ and re-normalize it. Let $\phi(t)$ denote the PDF of $\mathcal{N}(\mu, \sigma^2)$, and $\Phi(t)$ the CDF. Let

$$a = \frac{0 - \mu}{\sigma}, \quad b = \frac{1 - \mu}{\sigma}, \quad Z = \Phi(b) - \Phi(a).$$

Then, the truncated PDF function is equal to

$$f(t) = \frac{\phi\big(\frac{t-\mu}{\sigma}\big)}{\sigma Z},$$

where $t \in [0,1]$. The CDF is then equal to

$$F(t) = \frac{\Phi\big(\frac{t-\mu}{\sigma}\big) - \Phi(a))}{Z}$$

for $t \in [0,1]$. Assuming that $\mu \in [0,1]$, we get that the maximum density occurs at $\phi(0)$, where

$$c = \sup_t f(t) = \frac{\phi(0)}{\sigma Z} = \frac{1}{\sigma Z \sqrt{2\pi}}.$$

Given a budget $K$, the threshold $\tau_K$ is the value satisfying $F(\tau_K) = 1 - K/M$, and hence the value satisfying

$$\Phi\Big(\frac{\tau_K - \mu}{\sigma}\Big) - \Phi(b) - \alpha Z.$$

Hence, $\tau_K = \mu + \sigma \cdot \Phi^{-1}(\Phi(b) - \alpha Z)$. Lastly, we compute $V_{\mathcal{U}^*}^K$ Let $y_K = (\tau_K - \mu)/\sigma$.

$$\begin{aligned}
V_{\mathcal{U}^*} = \int_{\tau_K}^1 t f(t) dt &= \frac{1}{Z} \int_{\tau_K}^1 t \frac{1}{\sigma} \phi\Big(\frac{t - \mu}{\sigma}\Big) \\
&= \frac{1}{Z} \int_{y_K}^b (\mu + \sigma y) \phi(y) dy \\
&= \frac{1}{Z} \Big(\mu \int_{y_K}^b \phi(y) dy + \sigma \int_{y_K}^b y \phi(y) dy\Big) \\
&= \frac{\mu \cdot (\Phi(b) - \Phi(y_K)) + \sigma \cdot (\phi(y_K) - \phi(b))}{Z},
\end{aligned}$$

where we used the change of variable $y = \frac{t-\mu}{\sigma}$, and so $t = \mu + \sigma y$ and $dt = \sigma dy$.

For example, for $\mu = 0.5, \sigma = 0.15$, and budget $K/M = 0.5$, we get $\tau_K \approx 0.5$, $V_{\mathcal{U}^*} \approx 0.31$, $c \approx 2.66$, and $\gamma \leq 0.11$. For $\mu = 0.3, \sigma = 0.2$, and budget $K/M = 0.75$, we get $\tau_K \approx 0.19$, $V_{\mathcal{U}^*} \approx 0.23$, $c \approx 2.13$, and $\gamma \leq 0.13$. For $\mu = 0.7, \sigma = 0.10$, and budget $K/M = 0.25$, we get $\tau_K \approx 0.77$, $V_{\mathcal{U}^*} \approx 0.19$, $c \approx 3.98$, and $\gamma \leq 0.07$.

## C.2 General CDF condition

**Claim 4.1.** *For any budget $K \leq M$, Algorithm 1 called with parameters $\epsilon, \delta, \rho$ returns a $(1 - \epsilon, \delta)$-OPT allocation if and only if the distribution of treatment effect values satisfies:*

$$\int_{\tau_K}^{\tau_K + 2\rho} t f(t)\, dt - \int_{\tau_K - 2\rho}^{\tau_K - \alpha_K} t f(t)\, dt\, dt \leq \epsilon \int_{\tau_K}^1 t f(t). \tag{1}$$

*Proof.* By Claims B.7 and B.8, we know that

$$\frac{V_{\mathcal{U}_{\text{LEA}}}([0,1])}{V_{\mathcal{U}^*}([0,1])} = \frac{V_{\mathcal{U}_{\text{LEA}}}(A_1) + V_{\mathcal{U}_{\text{LEA}}^0}([0,1])}{V_{\mathcal{U}^*}(A_1) + V_{\mathcal{U}^*}(D)} = \frac{V(A_1) + V_{\mathcal{U}_{\text{LEA}}^0}}{V(A_1) + V_{\mathcal{U}^*}(D)}.$$

By re-arranging the expression in a manner similar to Claim 3.1 (but without using lower and upper bounds), we get that:

$$\frac{V_{\mathcal{U}_{\text{LEA}}}([0,1])}{V_{\mathcal{U}^*}([0,1])} = \frac{V(A_1) + V_{\mathcal{U}^*}(D) - V_{\mathcal{U}^*}(D) + V_{\mathcal{U}_{\text{LEA}}^0}}{V(A_1) + V_{\mathcal{U}^*}(D)} = 1 - \frac{V_{\mathcal{U}^*}(D) - V_{\mathcal{U}_{\text{LEA}}^0}}{V(A_1) + V_{\mathcal{U}^*}(D)}.$$

By definition of the value function and the PDF $f(t)$, it follows that

$$V(A_1) = \int_{\tau_K + 2\rho}^{1} t f(t) dt, \qquad V_{\mathcal{U}^*}(D) = \int_{\tau_K}^{\tau_K + 2\rho} t f(t) dt.$$

As for $V_{\mathcal{U}_{\text{LEA}}^0}$, by Claim 5 it follows that all units $u \in \mathcal{U}_{\text{LEA}}$ satisfy $\tau(u) \geq \tau_K - 2\rho$. Therefore, in the worst case, Algorithm 1 selects the $K_0$ units with the lowest $\tau(u)$ values in the interval $D = [\tau_K, \tau_K + 2\rho]$. This is precisely what the definition of $\alpha_K$ captures (Definition 10), and so it follows that

$$V_{\mathcal{U}_{\text{LEA}}^0} \geq \int_{\tau_K - 2\rho}^{\tau_K - \alpha_K} t f(t) dt,$$

with the equality holding in the worst-case instance of Algorithm 1 (even with high probability). Hence, we get a $(1 - \epsilon, \delta)$-OPT allocation whenever

$$\frac{\int_{\tau_K}^{\tau_K + 2\rho} t f(t) dt - \int_{\tau_K - 2\rho}^{\tau_K - \alpha_K} t f(t) dt}{\int_{\tau_K}^{1} t f(t) dt} \leq \epsilon,$$

as we wanted to show. $\square$

We can use our low-accuracy estimates $\hat{\tau}_K$ to check whether Equation 1 is satisfied (note that this is only a one-sided guarantee; failing to satisfy it does not imply that the allocation achieved by Algorithm 1 is not optimal, given that we use various upper bounds).

To do so, we want to compute an upper bound of the expression

$$\frac{V_{\mathcal{U}^*}(D) - V_{\mathcal{U}_{\text{LEA}}^0}}{V_{\mathcal{U}^*}([0,1])} \tag{4}$$

using our $\hat{\tau}$ estimates. To lower bound the denominator, the number of units that have $\tau(u) \in [\tau_K, 1]$ is lower-bounded by $|u : \hat{\tau}(u) \in [\tau_K + \rho, 1]|$. For each of the $\hat{\tau}(u)$ values that we find in this interval, by $\rho$-accuracy we know that $\tau(u) \geq \hat{\tau}(u) - \rho$. Hence,

$$V_{\mathcal{U}^*}([0,1]) \geq \sum_{u : \hat{\tau}(u) \geq \tau_K + \rho} \hat{\tau}(u) - \rho.$$

Because we do not have access to the true $\tau_K$, but we know that $|\hat{\tau}_K - \tau_K| \leq \rho$ by Claim 5, we can actually compute the lower bound as

$$V_{\mathcal{U}^*}([0,1]) \geq \sum_{u : \hat{\tau}(u) \geq \hat{\tau}_K + 2\rho} \hat{\tau}(u) - \rho. \tag{5}$$

Next, we upper bound $V(D)$. The number of units that have $\tau(u) \in [\tau_K, \tau_K + 2\rho]$ is upper-bounded by $|u : \hat{\tau}(u) \in [\tau_K - \rho, \tau_K + 3\rho]|$. For each of the $\hat{\tau}(u)$ values that we find in this interval, by $\rho$-accuracy we know that $\tau(u) \leq \hat{\tau}(u) + \rho$. Hence,

$$V_{\mathcal{U}^*}(D) \leq \sum_{u : \hat{\tau}(u) \in [\tau_K - \rho, \tau_K + 3\rho]} \hat{\tau}(u) + \rho.$$

Again because we do not have access to the true $\tau_K$ but we do have the guarantee that $|\hat{\tau}_K - \tau_K| \leq \rho$, it follows that

$$V_{\mathcal{U}^*}(D) \leq \sum_{u : \hat{\tau}(u) \in [\hat{\tau}_K - 2\rho, \hat{\tau}_K + 4\rho]} \hat{\tau}(u) + \rho. \tag{6}$$

Lastly, as for $V_{\mathcal{U}_{\mathsf{LEA}}^0}$, we are interested in lower bounding the number of units that belong to $\mathcal{U}_{\mathsf{LEA}}^0$. To do so, first we lower bound the quantity $K_0$, which corresponds to upper bounding the quantity $K_1$. We know that $K_1 \leq |u : \hat{\tau} \in [\tau_K + \rho, 1]| \leq |u : \hat{\tau} \in [\hat{\tau}_K, 1]|$. Hence, $K_0 \geq K - |u : \hat{\tau} \in [\hat{\tau}_K, 1]|$. Recall that all units in $\mathcal{U}_{\mathsf{LEA}}$ satisfy $\tau(u) \geq \tau_K - 2\rho$. To lower bound the value of $V_{\mathcal{U}_{\mathsf{LEA}}^0}$, we add up the first $K - |u : \hat{\tau} \in [\hat{\tau}_K, 1]|$ units that have $\hat{\tau}(u)$ value above $\hat{\tau}_K - 3\rho$, in increasing order.

Putting the three terms together, that is, our bounds for $V_{\mathcal{U}^*}([0,1])$ (Equation 5), $V_{\mathcal{U}^*}(D)$ (Equation 6), and $V_{\mathcal{U}_{\mathsf{LEA}}^0}$ (the sum of the first $K - |u : \hat{\tau} \in [\hat{\tau}_K, 1]|$ units that have $\hat{\tau}(u)$ value above $\hat{\tau}_K - 3\rho$) computed with our low-approximation $\hat{\tau}$ values, we have obtained an upper bound to the expression in Equation 4 and thus to Equation 1 in Claim 1.

Hence, if our upper bound is smaller than $\epsilon$, we have a guarantee that Algorithm 1 has returned an $(1 - \epsilon, \delta)$-OPT approximation.

It is clear that, definitionally, we get a $(1 - \epsilon)$-OPT allocation whenever we satisfy the inequality $\frac{V_{\mathcal{U}_{\mathsf{LEA}}}([0,1])}{V_{\mathcal{U}^*}([0,1])} \geq 1 - \epsilon$. The importance of Lemma 4.1 is that it shows how we can check (one sided) whether we have obtained an optimal allocation using only our $\hat{\tau}_K$ estimates.

We give an example of how to apply Claim 4.1 for a specific distribution by repeating the accuracy and sample complexity calculation for the case where the CATE $\tau$ values are uniformly distributed, this time with a tight accuracy analysis rather than using lower and upper bounds for $V_{A_2}(I)$ and $V_{A_2}(\mathsf{ALLOC}^*)$, respectively.

**Tight accuracy calculation for the uniform distribution.** Let the treatment effect values be distributed as $\mathrm{Unif}(0, 1)$. Then, the PDF corresponds to the function $f(t) = 1$. We compute each of the three terms in the RHS of Equation 1. Given that the uniform distribution is symmetric around $\tau_K$ for any value of $\tau_K$, it follows that $\alpha_K = 0$ for all $K$.

$$V_{\mathcal{U}^*}([0,1]) = \sum_{u:\tau(u)\in[\tau_K,1]} \tau(u) = \int_{\tau_K}^{\tau_K+2\rho} t\,dt = \left[\frac{t^2}{2}\right]_{\tau_K+2\rho}^{1} = \frac{1}{2} - \frac{(\tau_K)^2}{2} - 2\tau_K\rho - 2\rho^2.$$

$$V_{\mathcal{U}^*}(D) = \sum_{u:\tau(u)\in[\tau_K,\tau_K+2\rho]} \tau(u) = \int_{\tau_K}^{\tau_K+2\rho} t\,dt = \left[\frac{t^2}{2}\right]_{\tau_K}^{\tau_K+2\rho} = 2\tau_K\rho + 2\rho^2.$$

$$V_{\mathcal{U}_{\mathsf{LEA}}^0} = \sum_{u:\tau(u)\in[\tau_K-2\rho,\tau_K-\alpha_K]} \tau(u) = \int_{\tau_K-2\rho}^{\tau_K-\alpha_K} t\,dt = \left[\frac{t^2}{2}\right]_{\tau_K-2\rho}^{\tau_K} = 2\tau_K\rho - 2\rho^2,$$

where the equality holds in the worst-case high probability output of Algorithm 1; otherwise it is a lower bound. That is, the computation for $V_{\mathcal{U}_{\mathsf{LEA}}^0}$ corresponds to the worst-case output of Algorithm 1, as is the definition of $\alpha_K$. Then, $V_{\mathcal{U}^*}(D) - V_{\mathcal{U}_{\mathsf{LEA}}^0} = 4\rho^2$ and

$$\frac{V_{\mathcal{U}_{\mathsf{LEA}}}([0,1])}{V_{\mathcal{U}^*}([0,1])} = 1 - \frac{V_{\mathcal{U}^*}(D) - V_{\mathcal{U}_{\mathsf{LEA}}^0}}{V(A_1) + V_{\mathcal{U}^*}(D)} = 1 - \frac{V_{\mathcal{U}^*}(D) - V_{\mathcal{U}_{\mathsf{LEA}}^0}}{V_{\mathcal{U}^*}([0,1])} \geq 1 - \frac{4\rho^2}{(1-(\tau_K)^2)/2}.$$

Hence, by setting $\rho = \sqrt{\frac{\epsilon}{4}\left(\frac{1}{2} - \frac{(\tau_K)^2}{2}\right)}$, we obtain a $(1-\epsilon, \delta)$-OPT approximation of $\mathsf{ALLOC}^*$ with $O(M \ln(2M/\delta)/\epsilon)$ many samples.

**Remark C.1.** *These tight accuracy calculations assume a worst-case analysis (given the definition of the* value *of an allocation stated in Definition 6), rather than only achieving the looser notion of a $(1 - \epsilon)$-OPT approximation on expectation.*

## D  FLEXIBLE BUDGET

**Very dense distributions.** Consider the following "2 spikes" case, where $M/2$ units have $\tau(u)$ value equal to $1/2 - 2\epsilon$, and the other $M/2$ units have $\tau(u)$ value equal to $1/2 + 2\epsilon$. This example is connected to the usual lower-bound that appears in the sample complexity of bandit algorithms Bubeck et al. (2013); Kaufmann et al. (2016). We show that for this 2 spikes case, no budget is able to obtain a $(1 - \epsilon)$-optimal allocation.

For any budget $K \leq M/2$ we have that

$$V_{\mathcal{U}^*}([0,1]) = \frac{K}{2} \cdot \left( \frac{1}{2} + 2\epsilon \right) = \frac{K}{4} + \epsilon K.$$

As for the low-accuracy estimation algorithm, given that our $\hat{\tau}$ estimates are computed to $\rho = \gamma \sqrt{\epsilon}$ accuracy, the value of a random allocation, which we denote by $V_{\mathrm{rand}}$, upper-bounds the worst-case value realized by Algorithm 1.

On expectation, we have that

$$V_{\mathrm{rand}} = \frac{K}{2} \cdot \left( \frac{1}{2} - 2\epsilon \right) + \frac{K}{2} \cdot \left( \frac{1}{2} + 2\epsilon \right) = \frac{K}{2}.$$

The worst-case value is naturally given by $K(1/2 - 2\epsilon) = K/2 - 2\epsilon K$. By definition, an $(1 - \epsilon, \delta)$-OPT allocation $f$ must realize value

$$\begin{aligned} V_f([0,1]) \geq (1 - \epsilon)V_{\mathcal{U}^*}([0,1]) &= (1 - \epsilon) \cdot K \cdot \left( \frac{1}{2} + 2\epsilon \right) \\ &= \frac{K}{2} + 2\epsilon K - K \cdot \left( \frac{\epsilon}{2} + 2\epsilon^2 \right) \\ &= \frac{K}{2} + K \cdot \left( \frac{3\epsilon}{2} + 2\epsilon^2 \right) \end{aligned}$$

with probability at least $1 - \delta$. Hence, for any budget $K$, the expected value of a random allocation never yields a $(1 - \epsilon, \delta)$-optimal allocation, and is always short of at least $K(3\epsilon/2 + 2\epsilon^2)$ much value. Thus, the worst-case of Algorithm 1 fails to satisfy the $(1 - \epsilon, \delta)$-optimality requirement as well.

Nonetheless, we are quite close to achieving the value required by an optimal allocation. This presents an interesting duality for the problem of allocation: either the distribution of treatment effect values is $\rho$-regular, in which case Algorithm 1 returns an optimal allocation, or the distribution is very dense around the threshold, in which case Algorithm 1 (and even a random allocation) still does quite well. In order to realize the required value, we can either promise a $(1 - \kappa\epsilon)$-OPT allocation, or we can increase the budget in order to include more units. This type of overspending/resource augmentation is different from the modification of the budget discussed at the beginning of this section (which cannot work in this 2 spikes case, as we just showed). Here, we add more units to our allocation $\mathcal{U}_{\mathrm{LEA}}$, but we compete against the *original* budget $K$.

For the former, we want to find the smallest $\kappa$ such that a random allocation realizes an $(1 - \kappa\epsilon)$-OPT allocation. This has value:

$$(1 - \kappa\epsilon)V_{\mathcal{U}^*}([0,1]) = (1 - \kappa\epsilon)\left( \frac{1}{2} + 2\epsilon \right) \cdot K = \frac{K}{2} + \frac{4\epsilon K - \kappa\epsilon K}{2} - 2\kappa\epsilon^2 K.$$

Given that the expected value of $V_{\mathrm{rand}}([0,1])$ is $K/2$, we need $\kappa$ to satisfy

$$\frac{4\epsilon K - \kappa\epsilon K}{2} = 0 \implies \kappa \geq 4.$$

Indeed, a $(1 - 4\epsilon)$-OPT allocation requires realizing value at least

$$(1 - 4\epsilon)V_{\mathcal{U}^*}([0,1]) = (1 - 4\epsilon)\left( \frac{1}{2} + 2\epsilon \right) \cdot K = \left( \frac{1}{2} - 2\epsilon + 2\epsilon - 8\epsilon^2 \right) \cdot K = \frac{K}{2} - 8\epsilon^2 K,$$

and so $V_{\mathrm{rand}}([0,1]) \geq (1 - 4\epsilon)V_{\mathcal{U}^*}([0,1])$, as desired. Performing a similar calculation with the worst-case scenario of Algorithm 1, we get $\kappa \geq 8$.

**Overspending.** Alternatively, we can take the dual approach and ask what is the smallest budget $K' > K$ that allows a random allocation to get at least a $(1 - \epsilon)$-OPT allocation, where optimality is measured with respect to the original budget $K$. This is a form of resource augmentation. For any budget $K' > K$, we showed that $V_{\mathrm{rand}}([0,1]) = K'/2$. We want a $K'$ that satisfies

$$\frac{K'}{2} \geq (1 - \epsilon)V_{\mathcal{U}^*}([0,1]) = (1 - \epsilon) \cdot K \cdot \left( \frac{1}{2} + 2\epsilon \right).$$

This means that $K'$ must satisfy

$$K' \geq K + 4\epsilon K - \epsilon K - 4\epsilon^2 K = K + 3\epsilon K - 4\epsilon^2 K.$$

Hence, we need $K' \geq K + 3\epsilon K = K(1 + 3\epsilon)$.

We can apply this overspending/resource augmentation idea to a distribution that is not $\rho$-regular around the initial budget threshold $\tau_K$. That is, suppose that the initial budget $K$ is such that the interval $[\tau_K - 2\rho, \tau_K + 2\rho]$ contains a lot of probability mass. What is the minimum number $S$ of extra units that we need to spend on such that we achieve a $(1-\epsilon)$-OPT allocation, where optimality is measured with respect to the original budget $K$? The extra units that we add to $\mathcal{U}_{\mathsf{LEA}}$ are naturally added in descending order of $\hat{\tau}$ value starting at $\hat{\tau}_K$, and we denote their added value by $V_{\mathcal{U}^S_{\mathrm{extra}}}$. That is, the final threshold is $\tau_{K+S}$. Because we assume that the interval $[\tau_K - 2\rho, \tau_K + 2\rho]$ is highly dense, we lower bound the value $\hat{\tau}(u)$ of all of the extra units that we add by $\tau_K - 2\rho$. Then:

$$
\begin{aligned}
\frac{V^{K+S}_{\mathcal{U}_{\mathsf{LEA}}}([0,1])}{V^K_{\mathcal{U}}([0,1])} &= \frac{V(A_1) + V_{\mathcal{U}^0_{\mathsf{LEA}}} + V_{\mathcal{U}^S_{\mathrm{extra}}}}{V(A_1) + V_{\mathcal{U}^*}(D)} \\
&= 1 - \frac{V_{\mathcal{U}^*}(D) - V_{\mathcal{U}^0_{\mathsf{LEA}}} - V_{\mathcal{U}^S_{\mathrm{extra}}}}{V(A_1) + V_{\mathcal{U}^*}(D)} \\
&\geq \frac{(\tau_K + 2\rho)K_0 - (\tau_K - 2\rho)K_0 - V_{\mathcal{U}^S_{\mathrm{extra}}}}{V(A_1) + (\tau_K + 2\rho)K_0} \\
&\geq \frac{(\tau_K + 2\rho)K_0 - (\tau_K - 2\rho)K_0 - (\tau_K - 2\rho)S}{V(A_1) + (\tau_K + 2\rho)K_0} \\
&= \frac{4\rho K_0 - (\tau_K - 2\rho)S}{V(A_1) + (\tau_K + 2\rho)K_0}.
\end{aligned}
$$

By setting

$$\frac{4\rho K_0 - (\tau_K - 2\rho)S}{V(A_1) + (\tau_K + 2\rho)K_0} \leq \epsilon,$$

we get that the minimum number of units that we need to add to our initial budget $K$ is

$$S \geq \frac{\left(4\rho - \epsilon(\tau_K + 2\rho)\right)K_0 - \epsilon V(A_1)}{\tau_K - 2\rho}.$$

Asymptotically, this means that $S$ is of the order of $\sqrt{\epsilon}K_0$. As we will see in Section 6, in practice we only require adding one extra unit in order to realize the value of the optimal allocation.

**Choosing an appropriate budget: guidelines for policy-makers.** In the cases where the policy-maker has some flexibility over the budget, it is a sensible approach to refine the budget after running our algorithm. Choosing a budget $K$ such that $\tau_K$ lies in an interval (of width $\approx 2\rho$) of uniform-like probability mass is desirable for several reasons. First, it ensures that an optimal allocation is achieved, given that $\rho$-regularity is satisfied in the $D$-interval. Second, it ensures that the allocation is less arbitrary, as it minimizes the number of units that are very close to the threshold that determines whether or not treatment is given. As shown in Section 4.2, the low-accuracy estimates $\hat{\tau}$ give an approximation of the CDF curve as well, and so we can upper bound the probability mass contained in an interval $[a, b]$ by counting the number of units $u$ that have $\hat{\tau}(u)$ value in the interval $[a-\rho, b+\rho]$. If the interval is too dense, the policy-maker can slide the budget up or down in order to find a threshold $\hat{\tau}_K$ that lies in a low-mass interval.

While the 2 spikes case demonstrates that we cannot always hope to find such an interval, this only occurs in the cases where the CDF satisfies a strong "anti-Lipschitz" condition, where changing $K$ for a much larger or much smaller $K'$ induces very little difference between $\tau_K$ and $\tau_{K'}$. However, it is rare to find such densely-packed distributions in real-world RCTs, and it would be highly arbitrary to decide on an allocation in the case of these distributions. As we will see in Section 6 in our experiments for real-world data, we can always find a $K'$ very close to $K$ (even if we are only willing to underspend) in the few cases where Algorithm 1 does not yield an optimal allocation. Moreover, as we have show in this section, we can obtain optimal allocations for distributions with high mass around $\tau_K$ by overspending and adding some extra units to $\mathcal{U}_{\mathsf{LEA}}$. We also test this method in our experiments in Section 6.

Summarizing, if a budget does not work, the policy-maker can: (1) detect so correctly by using the approximation $\hat{F}$ of the CDF (this is a one-sided test), (2) slide the budget from $K$ to $K'$ as to find a smooth interval, or (3) if no interval is $\rho$-regular, then the distribution $\text{Pr}_\tau$ is very dense in the entire range, in which case Algorithm 1 should obtain a $(1 - \kappa\epsilon)$-OPT allocation for a small value of $\kappa$, or, equivalently, we can add a few extra units to $\mathcal{U}_{\text{LEA}}$ to realize the optimal value.

We test these strategies experimentally and report the results in Section F. We find that, in practice, we only need to slide the budget by at most 2 units in order to find a $K$ that obtains a $(1 - \epsilon)$-allocation with $O(M/\epsilon)$ many samples. As for the overspending/resource augmentation strategy, we find that in all cases adding one extra unit (namely, the unit corresponding to $\hat{\tau}_{K+1}$) suffices to realize a $(1 - \epsilon)$-optimal allocation.

## E  EXPERIMENTAL DETAILS

### E.1  RCT DATA

We test our algorithm on real-world RCT data in order to test its efficacy and study the $\rho$-regularity smoothness condition in practice. For robustness, we test our algorithm on data from five different real-world RCTs, which have been conducted across multiple domains. These datasets have recently been analyzed in the work of Sharma & Wilder (2025), and tested for different policies, which is why we choose these five datasets. We summarize each RCT dataset, following the descriptions given in Sharma & Wilder (2025) and in the original papers.

(1) *Tennessee's Student Teacher Achievement Ratio (STAR) project* (Pate-Bain et al., 1997). Domain: education. This is a 4-year RCT conducted by the Tennessee State Department of Education to examine class size effects on student performance. The study involved 11,601 students across 79 schools. Students and teachers were randomly assigned to class types beginning in kindergarten and followed through grade 3. The original study considered three possible treatments: small class ($\approx$13–17 students), regular class ($\approx$22–25 students), or regular class with a full-time aide. Following the analysis performed in Sharma & Wilder (2025), in order to keep the binary treatment/control study, treatment corresponds to having been assigned to a small class, whereas control corresponds to having been assigned to a regular class. The outcome is a cumulative test-score measure in kindergarten (which should be positively affected by the treatment), which is a composite outcome from four standardized test scores: reading, math, listening, and word skills.

We filter out out observations with missing test scores, treatment assignments, or school IDs, leaving a total of 3,712 students. Out of these, 1,989 are control and 1,723 are treated.

(2) *Targeting the Ultra Poor (TUP) in India* (Banerjee et al., 2021). Domain: economic development. This RCT studies the long-term effects of providing large one-time capital grants to low-income households. The treatment given was a one-time capital grant to ultra-poor households, and the control was to give no grant. The study measured how family income and overall consumption evolved over a period of 7 years. The outcome variable that we use in our experiments is the total household expenditure (which should be positively affected by the treatment). This is the difference between the endline consumption and the baseline consumption.

After filtering the dataset, we have 864 households, with 410 control and 454 treated.

(3) *National Supported Work (NSW) demonstration* (Dehejia & Wahba, 2002; 1999; LaLonde, 1986). Domain: labor economics. This study was carried out to analyze the impact of the National Supported Work Demonstration, which was a job training program, on the income of the participants in the year 1978. This job-training program targeting disadvantaged workers. The dataset also collected the participants' income in the year 1975 as a baseline. The treatment was assignment to the NSW training program, and the control was no assignment. The outcome variable that we use is the annual earnings in 1978 (which should be positively affected by the treatment).

After filtering the dataset, we have 722 individuals, out of which 425 are control and 297 are treatment.

(4) *Acupuncture dataset* (Vickers et al., 2004). Domain: healthcare. This RCT evaluates the effect of acupuncture therapy on patients with chronic headache, with assessments at randomization, 3 months, and 1 year. The treatment is accupuncture for chronic headaches, and control is the usual

care. The outcome variable that we use is the headache-severity score one year after randomization. Headache severity is measured using a discrete 0-5 Likert scale. Treatment is thus expected to lower the outcome, which is why we flip the sign in our figures for coherence with the other RCTs.

After filtering the dataset, we have 301 patients, with 140 control and 161 treatment.

(5) *Postoperative Pain dataset* (McHardy & Chung, 1999). Domain: Healthcare. This RCT investigates whether gargling a licorice solution prior to endotracheal intubation reduces postoperative sore throat, a common side effect of thoracic surgery using double-lumen tubes. Hence, treatment corresponds to receiving licorice gargle prior to endotracheal intubation, and control corresponds to placebo/standard care. The outcome variable we use is throat-pain score 4 hours post-surgery, measured on a on a discrete 0-7 Likert scale. As in the case of the acupuncture dataset, treatment is expected to lower the outcome, and so we also flip the sign in our figures for coherence with the other RCTs.

After filtering the dataset, we have 233 patients, with 116 control and 117 treatment.

## E.2    SEMI-SYNTHETIC EXPERIMENTS

**Grouping methods.**    For each of the five RCT datasets, we create a set of units $\mathcal{U}$ by partitioning the individuals into groups from their covariates. For each dataset, we describe the various grouping methods that we test in our algorithm. For any grouping strategy, we require at least 3 treated individuals and 3 control individuals, and that the treatment rate is between $15\%$ and $85\%$, to ensure balance within each unit. We only include the grouping methods that succeeded in creating a set of units satisfying these requirements, and discarded several other grouping methods based on the dataset variables.

(1) *STAR dataset*. We use the following grouping methods:

- *School groups.* We cluster the students by school ID.
- *Performance groups.* We group the students based on baseline performance percentiles (test score averages).
- *Causal forest groups.* We use random forest regressors to predict treatment effects (one for control and one for treatment), and then cluster the students based on predicted CATE values and covariates using $K$-means clustering. We cluster aiming for different number of groups (typically 30 and 50). We note that we use the name "causal forest" in the figures for short, but we are referring to CATE-based grouping obtained via random-forest and clustering; not through a causal forest estimator.
- *Propensity score groups.* We perform a stratification based on propensity scores using cross-validated logistic regression, dividing the propensity score distribution into equal-sized quantile-based strata (thus grouping students with similar propensity score).

(2) *TUP dataset*. We use the following grouping methods:

- *Baseline poverty groups.* We group the households using baseline consumption as a poverty proxy, clustering them using $K$-means.
- *Demographic groups.* We search for various demographic variables (such as "gender", "education", "age") and take their intersections.
- *Assets.* We identify asset-related variables (such as "land" or "livestock") and group them using $K$-means.
- *Causal forest groups.* We use random forest regressors to predict treatment effects, and then cluster the households based on the predicted CATE values and covariates using $K$-means.
- *Propensity score groups.* We perform a stratification based on propensity scores using cross-validated logistic regression.

(3) *NSW dataset*. We use the following grouping methods:

- *Baseline earnings groups.* We divide the employed individuals into percentile-based earnings brackets.

- *Age groups.* We create a percentile-based stratification of the age distribution.
- *Causal forest groups.* We use random forest regressors to predict treatment effects, and then cluster the individuals based on the predicted CATE values and covariates using $K$-means.
- *Propensity score groups.* We perform a stratification based on propensity scores using cross-validated logistic regression.

(4) *Acupuncture dataset.* We use the following grouping methods.

- *Age groups.* We create percentile-based age brackets.
- *Age-Chronicity interaction groups.* We compute a score for each patient combining their age and their chronicity variables, and then group them into equally-sized groups (in order of the scores).
- *Baseline headache.* We group the patients by their initial headache score.
- *Chronicity groups.* We group the patients by the length of chronic headache history.
- *Covariate forest groups.* We perform a $K$-means clustering based on the covariate profiles.
- *Multidimensional composite groups.* We use the variables age, chronicity, baseline headache score, and combine them into an average score.

(5) *Postoperative dataset.* We use the following grouping methods.

- *BMI groups.* We divide the patients into 30 equal-sized BMI brackets.
- *Age groups.* We perform a percentile-based stratification on the age variable.
- *Demographics.* We combine different preoperative patient characteristics.
- *Covariate forest.* We perform $K$-means clustering on the patient characteristics.

Note that for the acupuncture and postoperative dataset we have less data than for the other datasets, which is why the group sizes that we obtain are smaller and thus prone to higher error.

**Obtaining the CATE values.** Each grouping method for each dataset yields a partition of the dataset into a set $\mathcal{U}$ of $M$ units. For each unit, we compute $\tau(u)$ as follows: we compute the treated outcome mean (i.e., the mean of the outcome variable among the treated members of group $u$) and the control outcome mean (i.e., the mean of the outcome variable among the control members of group $u$). Subtracting the control outcome mean from the treatment outcome mean yields the treatment effect value for each unit. Lastly, we normalize all of the treatment effect values across the $M$ units into $[0, 1]$, yielding the final $\tau(u)$ values.

Once we have the $\tau(u)$ values, which allow us to compute the value of the optimal allocation, we simulate the sampling as follows. For every drawn sample of the population, we select one of the units $u$ uniformly at random and draw a sample from the distribution $\mathrm{Bern}(\tau(u))$. That is, each group represents an "arm" with a Bernoulli reward equal to its normalized CATE. In all experiments, we set the failure probability $\delta$ to 0.05.

### E.3 FIRST APPROACH: GENERATING FIGURE 1

For this approach, we do not select a specific value of $\epsilon$. Instead, we try different sample sizes, from $N = 100$ to $N = 20{,}000$ across different budget constraints. Specifically, we try budgets $K$ that are $10\%, 20\%, 30\%, 50\%, 70\%$, and $90\%$ of $M$. For each sample size and each budget, we sample from the population for that many samples and compute the estimates $\hat{\tau}(u)$ for each of the $M$ units $u$. Then, we select the $K$ units with the highest $\hat{\tau}$ estimates. Using the ground-truth values $\tau(u)$, we compute the value of this allocation. We then plot the realized value of the allocation ($y$-axis) for each sample size ($x$-axis), having one plot for each fixed budget and grouping method. We repeat this sampling process 50 times, obtaining an average value of the allocation for each sample size and confidence bounds.

In each figure, we also plot the value that we would expect from the worst-case bound of $O(M/\epsilon^2)$ and our bound of $O(M/\epsilon)$. Specifically, from Lemma 3, we have that the expected value of the

allocation using the worst-case bound is

$$\left(1 - \sqrt{\frac{M \ln(2M/\delta)}{N}}\right) \cdot V_{\mathcal{U}^*}.$$

From Theorem 7, we obtain that the expected value of the allocation using our theoretic bound for $\rho$-regular distributions is

$$\left(1 - \frac{M \ln(2M/\delta)}{N}\right) \cdot V_{\mathcal{U}^*}.$$

In Section F, we display these plots for each of the grouping methods for each of the datasets. For each, we plot the relationship between the normalized allocation value and the sample size for four choices of a budget.

### E.4 SECOND APPROACH: GENERATING FIGURE 2

For Figure 2 and its further displays in Section F (the last two plots for each subsection), we carry out our analysis differently.

Here, we do choose a value of $\epsilon$, which ranges from $0.001$ to $0.2$. Based on our computation of the value of $\gamma$ in Section C for various distributions, we choose to call Algorithm 1 with $\gamma = 0.5$. This bound is best decided using an informed guess of how the $\tau$ values are distributed; if $\gamma$ is too high then Algorithm 1 will have a higher error rate. For each value of $\epsilon$, we compute the corresponding number of samples in total that we use to construct our estimates $\hat{\tau}$ using our theoretical bound of $N = M \ln(2M/\delta)/\epsilon$. Note that each grouping method yields a fixed number of units $M$. Sampling that many units, we obtain the coarse estimates $\hat{\tau}(u)$, and for each possible budget $K \in \{1, \ldots, M\}$ we compute the value realized by our allocation (where we compute the value using the ground-truth estimates $\tau$, but they are selected using the low-accuracy estimates $\hat{\tau}$). This gives the value $V_{\mathcal{U}_{\mathsf{LEA}}}$ for each budget $K$.

Separately, for each possible budget $K \in \{1, \ldots, M\}$, we compute the value $V_{\mathcal{U}^*}$ of the optimal allocation. Then, we check whether or not $V_{\mathcal{U}_{\mathsf{LEA}}}/V_{\mathcal{U}^*} \geq 1 - \epsilon$, for each value of $K$ and for the fixed value of $\epsilon$. We compute the rate of failed budgets (i.e., the proportion out of $M$) that do not achieve a $(1 - \epsilon$-optimal allocation. We repeat this process 50 times for each combination of $K$ and $\epsilon$, giving us a robust average failure rate with confidence bounds. We do this for every grouping method and for every RCT dataset. The first row in Figure 2 plots some of these failure rates. The second-to-last plot in each subsection of Section F shows these plots for all of the grouping strategies for each RCT dataset. In all cases, we see that the average failure rate is below $5\%$. This is higher for the acupuncture and postoperative datasets, since we work with fewer groups in all grouping strategies (because the dataset is smaller). Moreover, we remark that in our experiments we test *all* budgets, and we do not discard budgets that are very small (e.g., $K = 1$). As per the $V_{\mathcal{U}^*}$ term in Theorem 8, very small values of $K$ increase the failure rate. The looser choice of $\gamma$ also increases the failure rate. For all these reasons, the failure rate that we obtain is higher than what it would be if we discarded small budgets and decreased $\gamma$ further, or if we had a higher number $M$ of units.

Second, we test our flexible budget strategies discussed in Section 5. In the cases where a budget $K$ fails to obtain a $(1 - \epsilon)$-optimal allocation (recall that we are always using the fewer $O(M/\epsilon)$ many samples to compute the allocation), we find the closest value of $K'$ such that we obtain a $(1 - \epsilon)$-optimal allocation, and then we report $|K - K'|$. We also compute such a $|K - K'|$ for $K' \leq K$; i.e., if we are only willing to underspend. We repeat this process 50 times for each $\epsilon$, grouping method, and RCT. Some of the results are reported in the second row of Figure 2; the comprehensive set of plots is in Section F. The last plot of each subsection of Section F shows these plots for all of the grouping strategies for each RCT dataset. In all cases, including the subcases where we only underspend, we see that, on average, $|K - K'| \leq 2$.

We also test our overspending idea of adding extra units to our allocation. We test it on each of the failed budget settings. In all cases, without exception, we find that adding one extra unit (i.e., the unit that corresponds to $\hat{\tau}_{K+1}$), realizes an optimal allocation.

The code for this paper can be found at: `https://github.com/silviacasac/alloc-vs-cate`.

*Use of LLMs.* We have used LLMs to aid in exploring the five RCTs and their various features, in order to find sensible grouping strategies for each RCT. After implementing the code for the base simulation of our algorithm, we used LLMs to help adapt the algorithm to the specifics of each dataset. Lastly, we have used LLMs to double-check the integrals computed for the Beta and Gaussian distributions.

## F  FURTHER EMPIRICAL RESULTS

### F.1  STAR DATASET

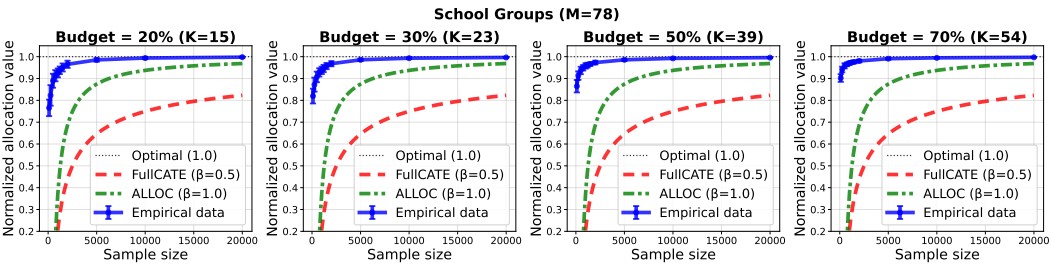

Figure 3: STAR dataset, school groups.

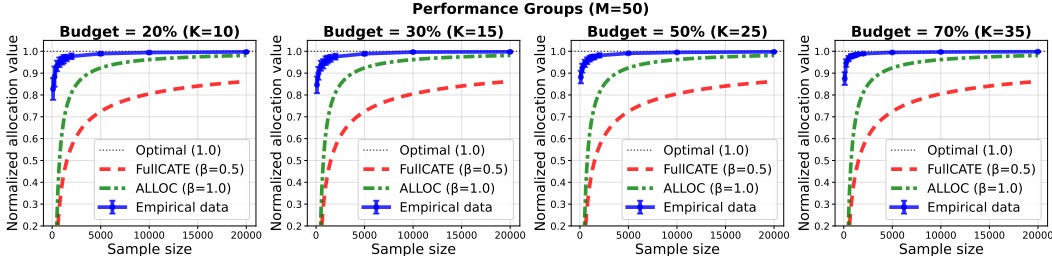

Figure 4: STAR dataset, performance groups.

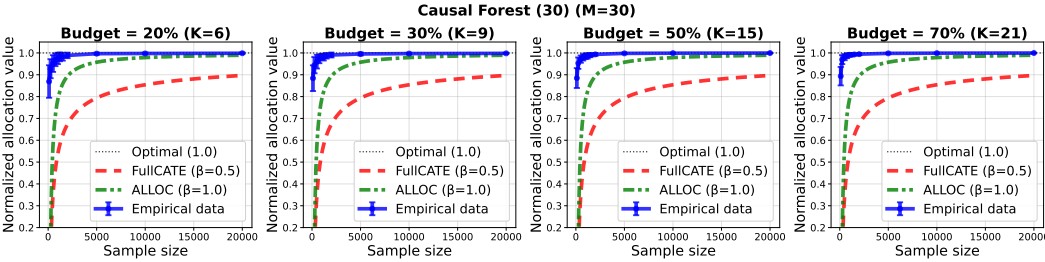

Figure 5: STAR dataset, causal forest 30.

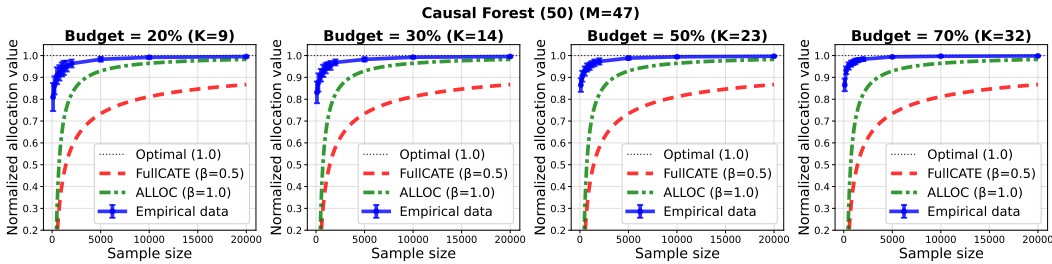

Figure 6: STAR dataset, causal forest 50.

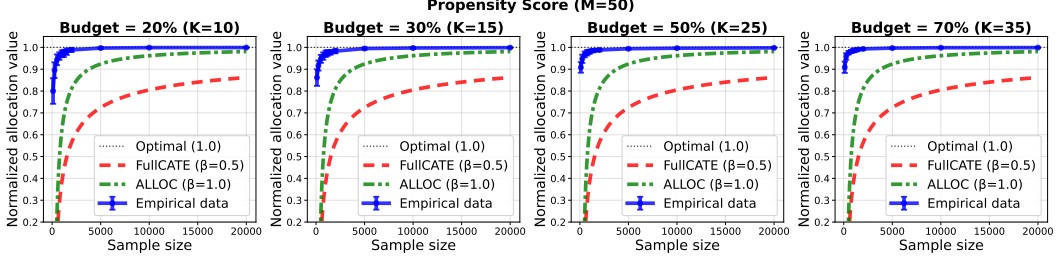

Figure 7: STAR dataset, propensity score.

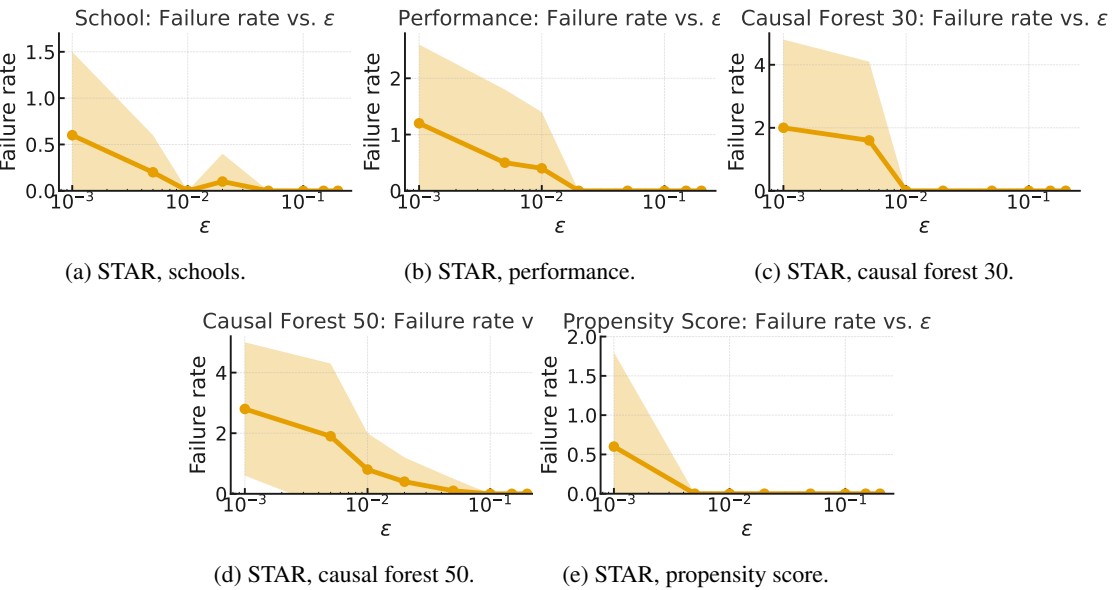

(a) STAR, schools.  (b) STAR, performance.  (c) STAR, causal forest 30.

(d) STAR, causal forest 50.  (e) STAR, propensity score.

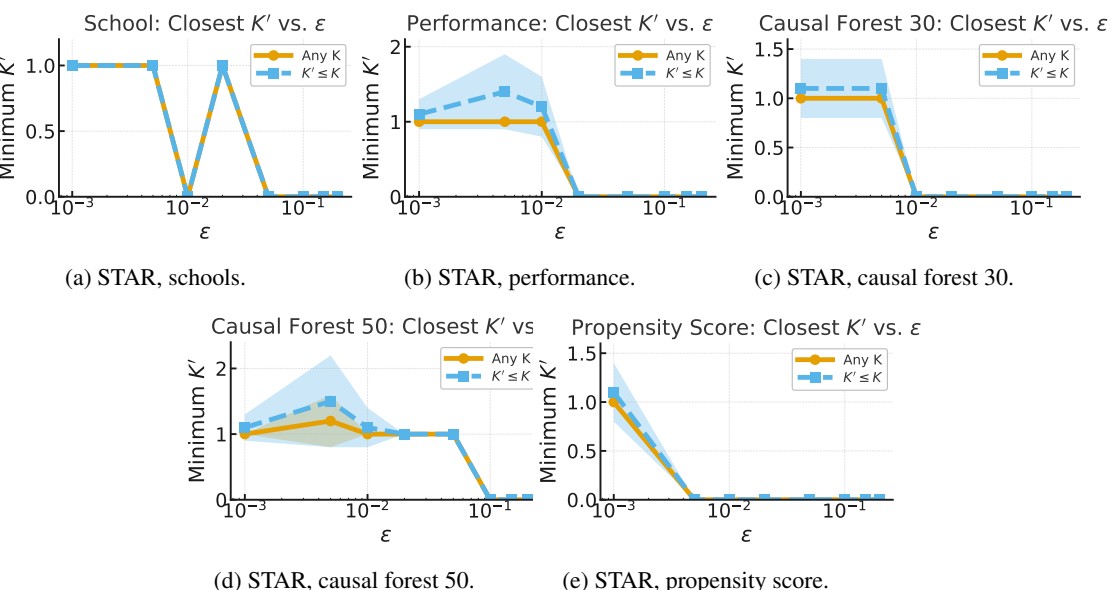

(a) STAR, schools.  (b) STAR, performance.  (c) STAR, causal forest 30.

(d) STAR, causal forest 50.  (e) STAR, propensity score.

## F.2 TUP DATASET

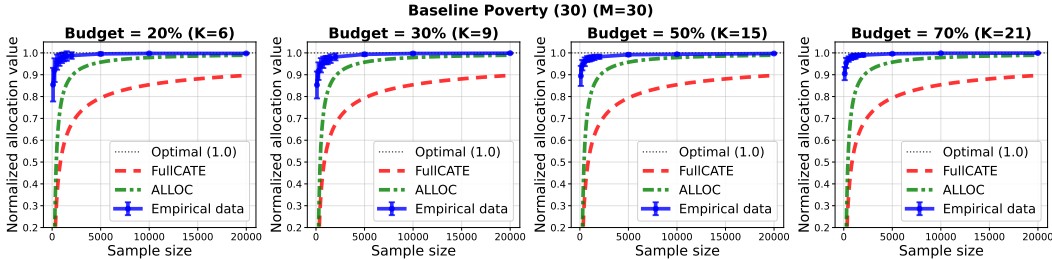

Figure 10: TUP dataset, baseline poverty 30.

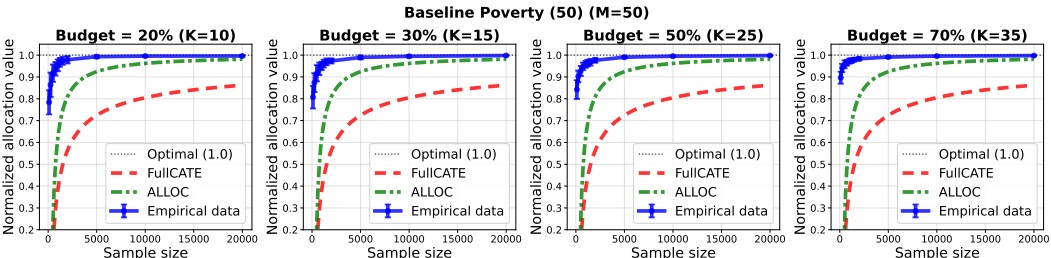

Figure 11: TUP dataset, baseline poverty 50.

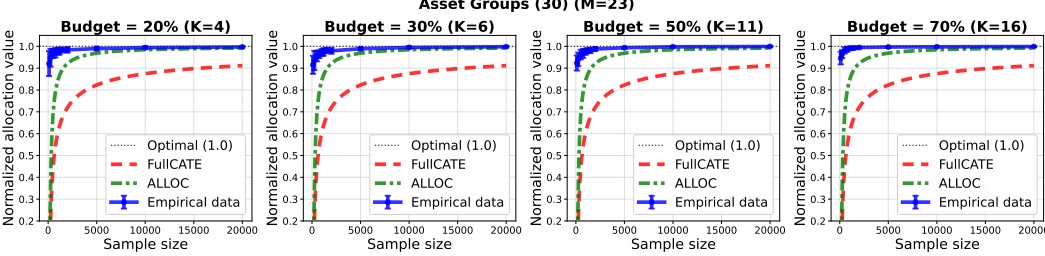

Figure 12: TUP dataset, asset groups 30.

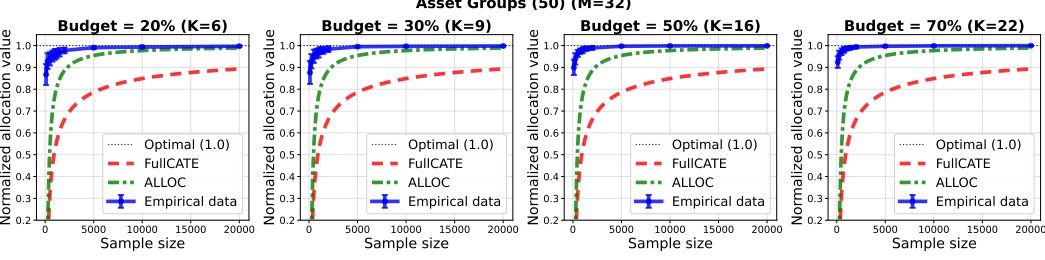

Figure 13: TUP dataset, asset groups 50.

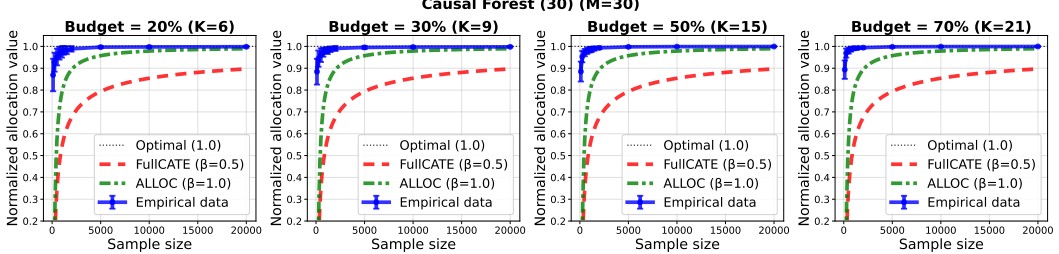

Figure 14: TUP dataset, causal forest 30.

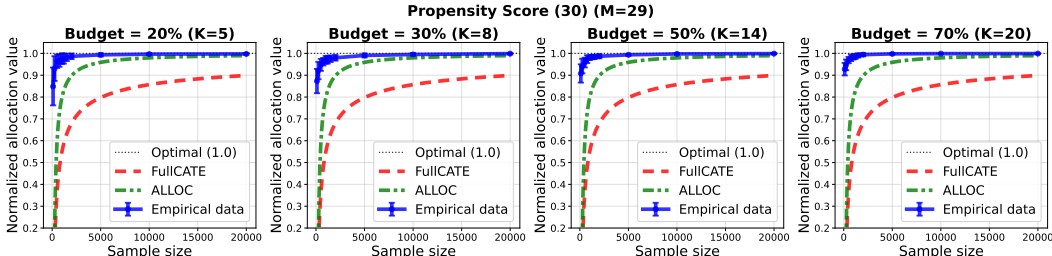

Figure 15: TUP dataset, propensity score.

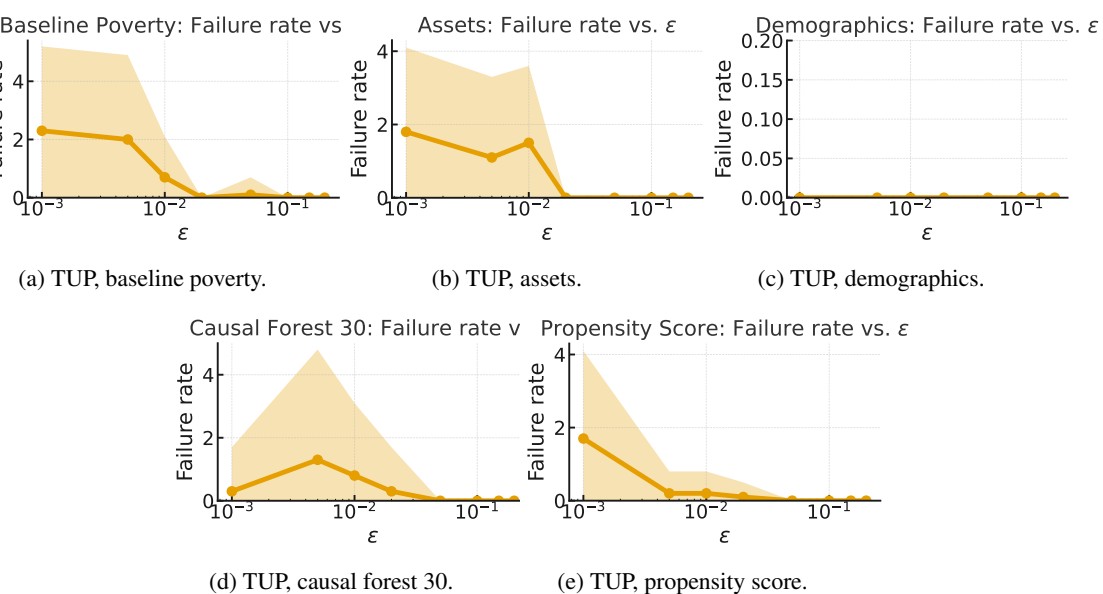

(a) TUP, baseline poverty.          (b) TUP, assets.          (c) TUP, demographics.

(d) TUP, causal forest 30.          (e) TUP, propensity score.

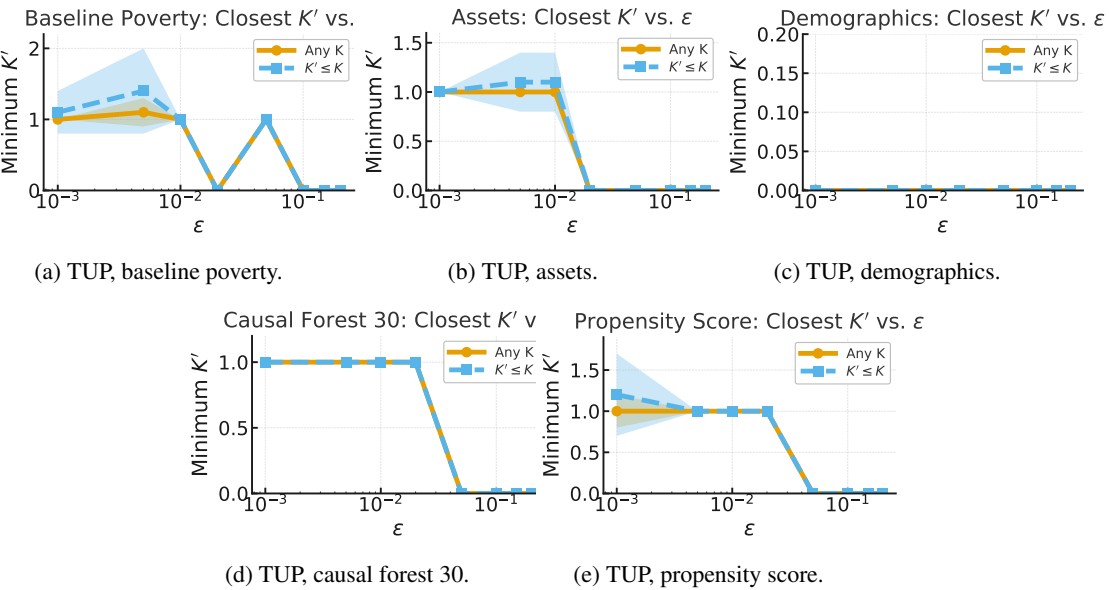

(a) TUP, baseline poverty.  (b) TUP, assets.  (c) TUP, demographics.

(d) TUP, causal forest 30.  (e) TUP, propensity score.

## F.3 NSW DATASET

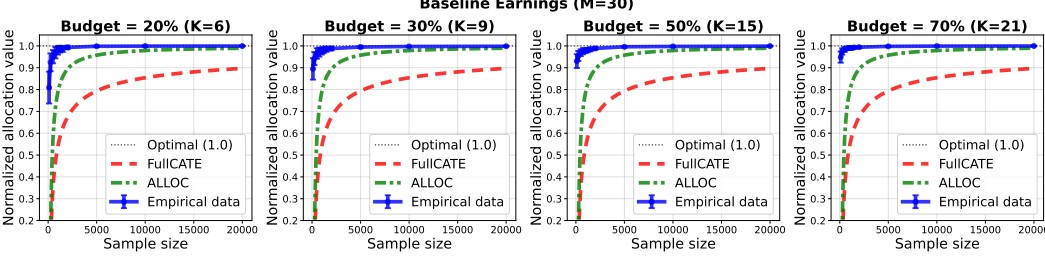

Figure 18: NSW dataset, baseline earnings.

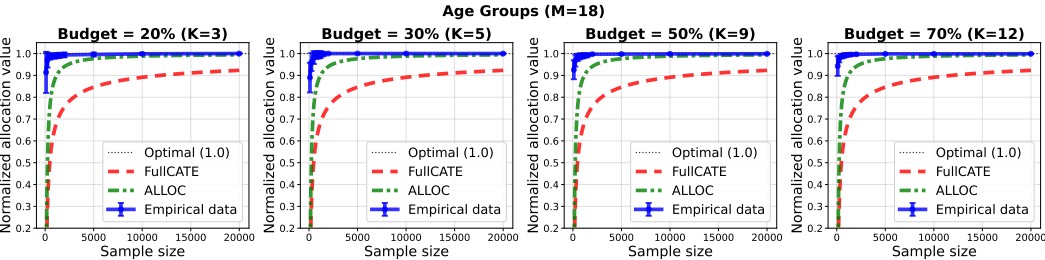

Figure 19: NSW dataset, age groups.

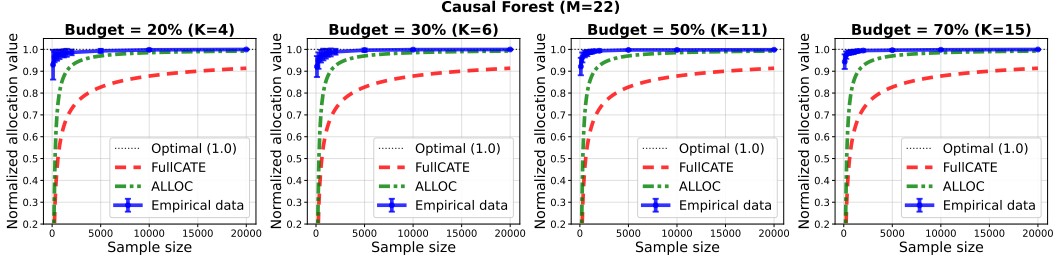

Figure 20: NSW dataset, causal forest.

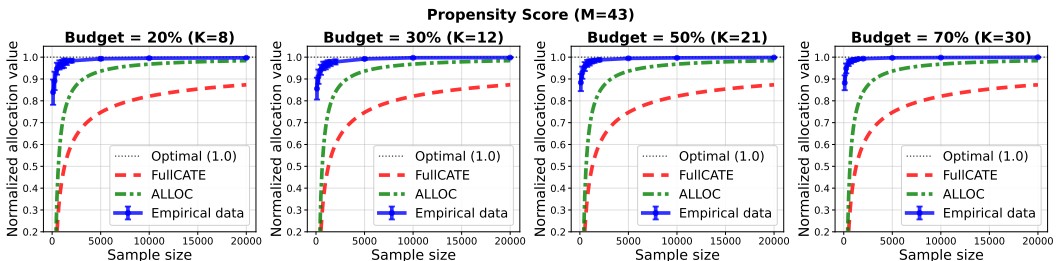

Figure 21: NSW dataset, propensity score.

## F.4 ACUPUNCTURE DATASET

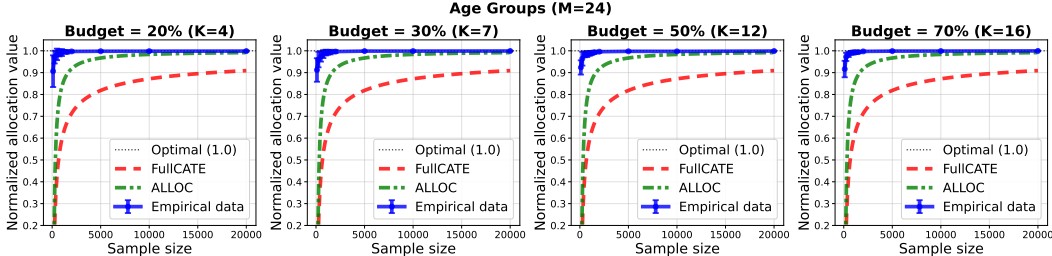

Figure 23: Acupuncture dataset, age groups.

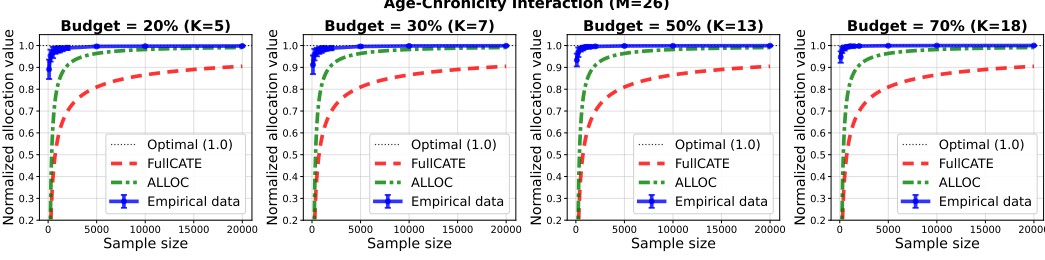

Figure 24: Acupuncture dataset, age-chronicity interaction.

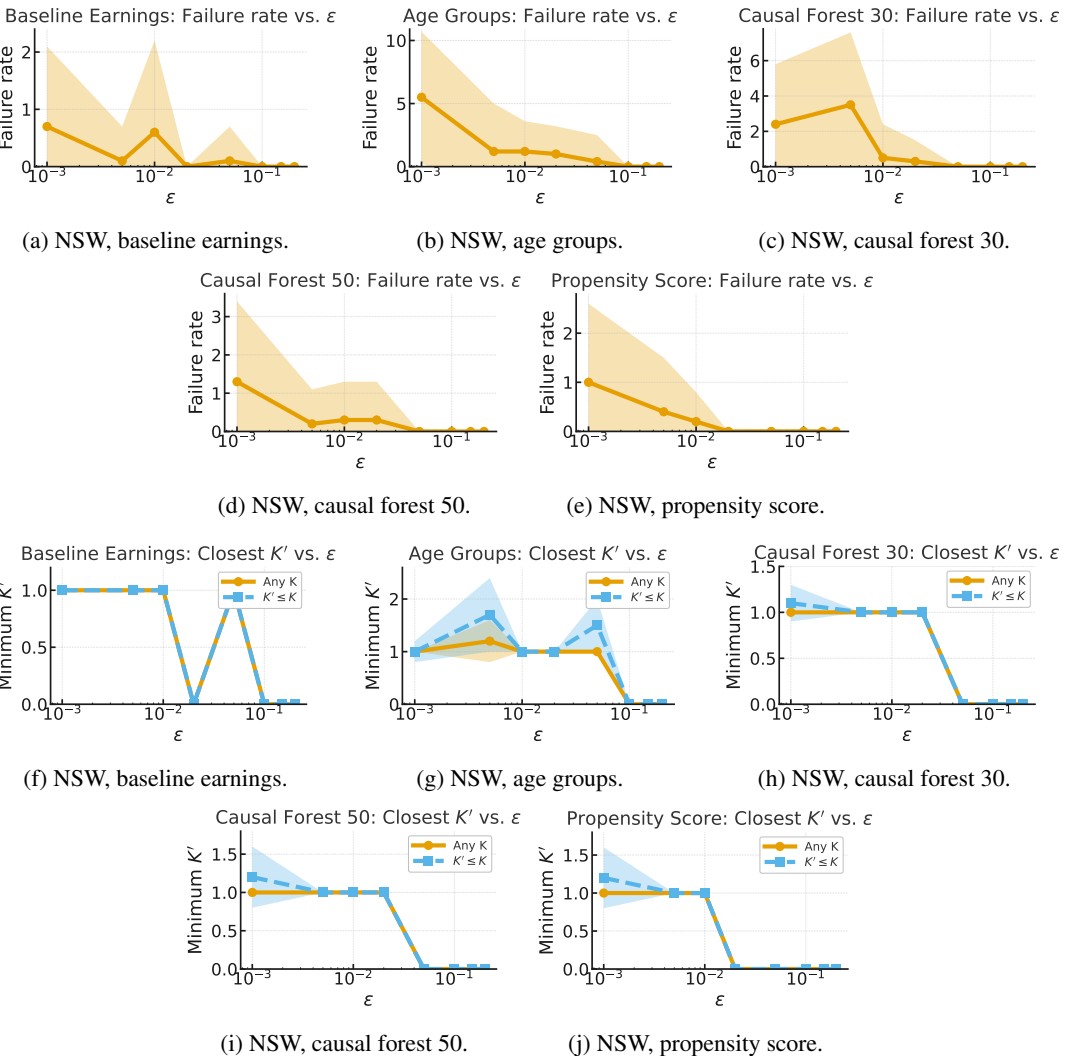

(a) NSW, baseline earnings.

(b) NSW, age groups.

(c) NSW, causal forest 30.

(d) NSW, causal forest 50.

(e) NSW, propensity score.

(f) NSW, baseline earnings.

(g) NSW, age groups.

(h) NSW, causal forest 30.

(i) NSW, causal forest 50.

(j) NSW, propensity score.

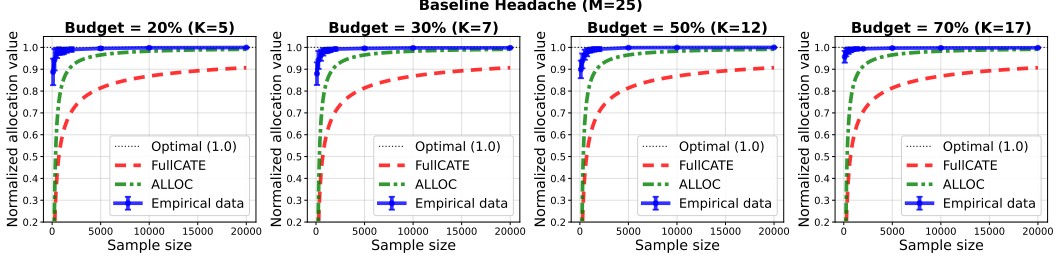

Figure 25: Acupuncture dataset, baseline headache.

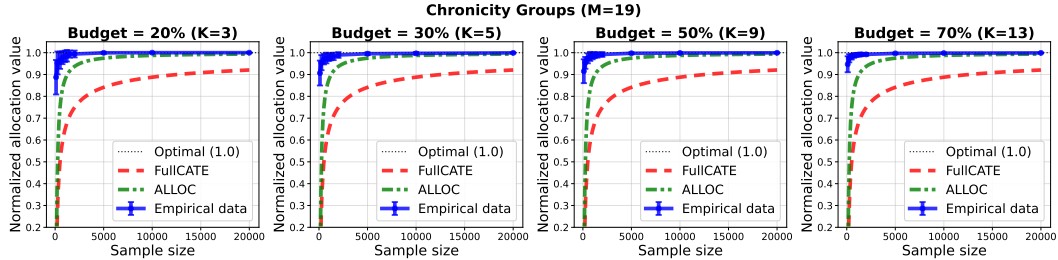

Figure 26: Acupuncture dataset, chronicity groups.

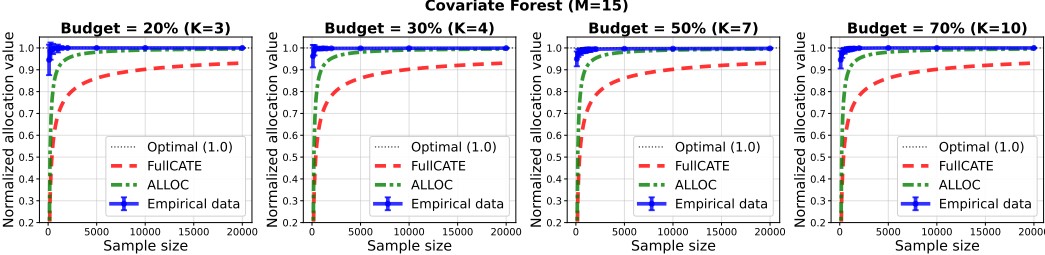

Figure 27: Acupuncture dataset, covariate forest.

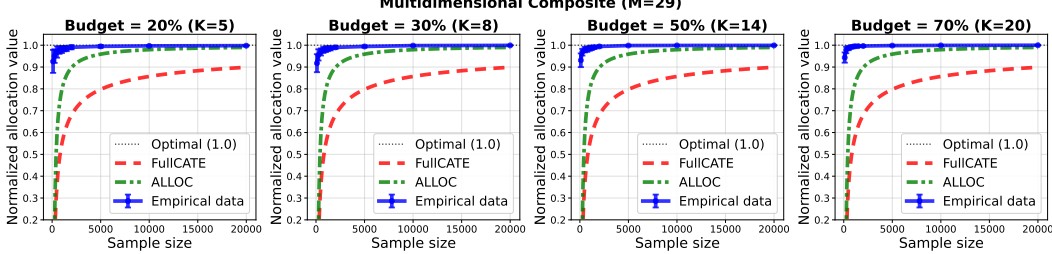

Figure 28: Acupuncture dataset, multidimensional.

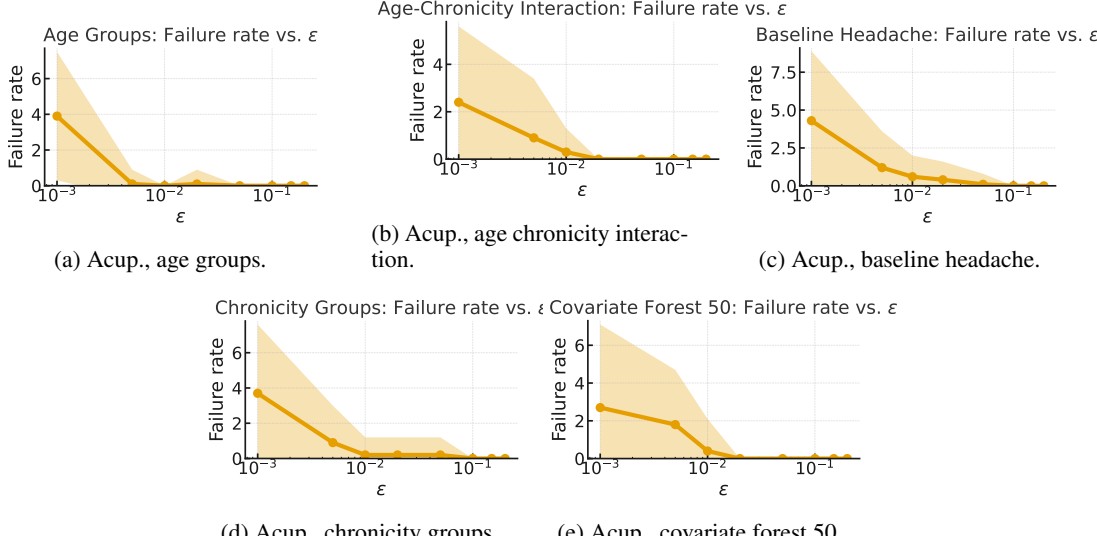

(a) Acup., age groups.

(b) Acup., age chronicity interaction.

(c) Acup., baseline headache.

(d) Acup., chronicity groups.

(e) Acup., covariate forest 50.

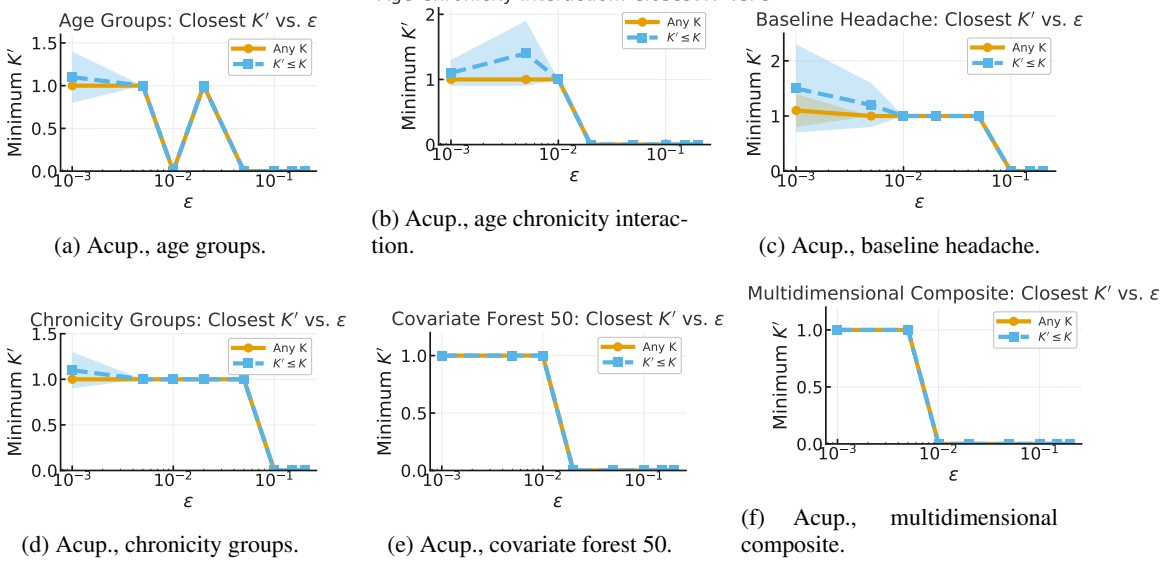

(a) Acup., age groups.

(b) Acup., age chronicity interaction.

(c) Acup., baseline headache.

(d) Acup., chronicity groups.

(e) Acup., covariate forest 50.

(f) Acup., multidimensional composite.

## F.5 POSTOPERATIVE DATASET

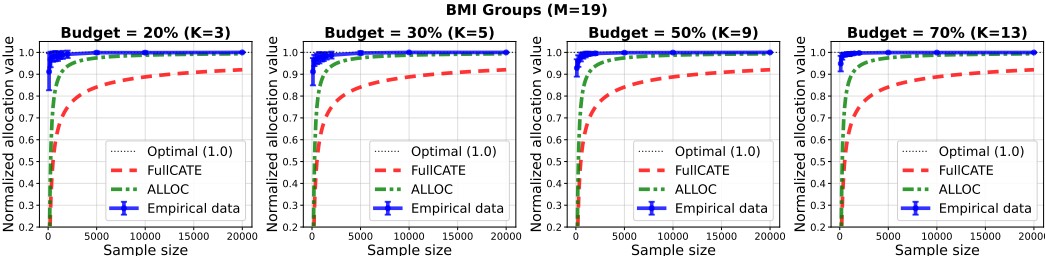

Figure 31: Postoperative dataset, BMI.

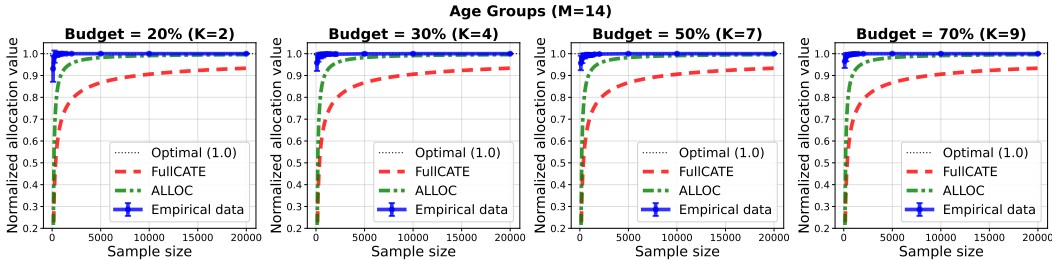

Figure 32: Postoperative dataset, age groups.

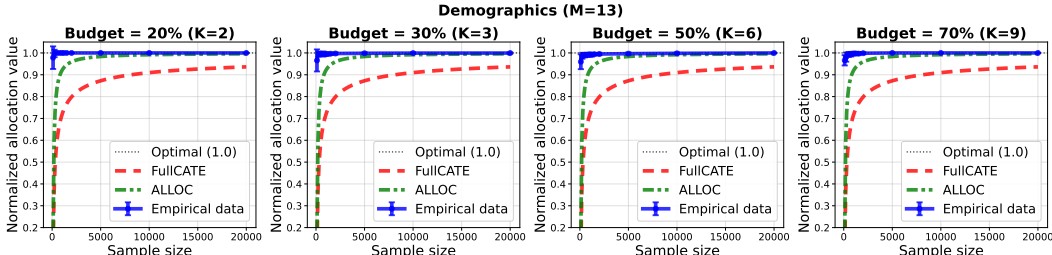

Figure 33: Postoperative dataset, demographics.

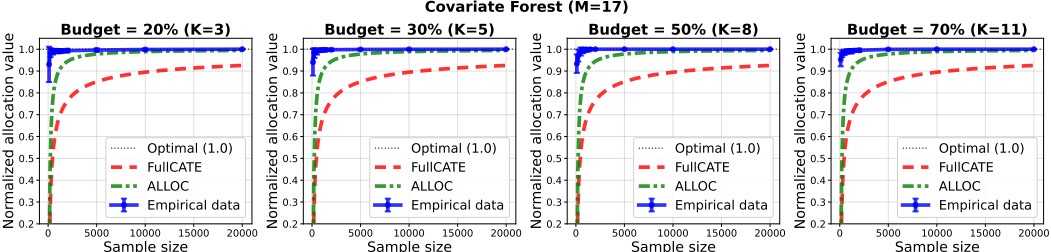

Figure 34: Postoperative dataset, covariate forest.

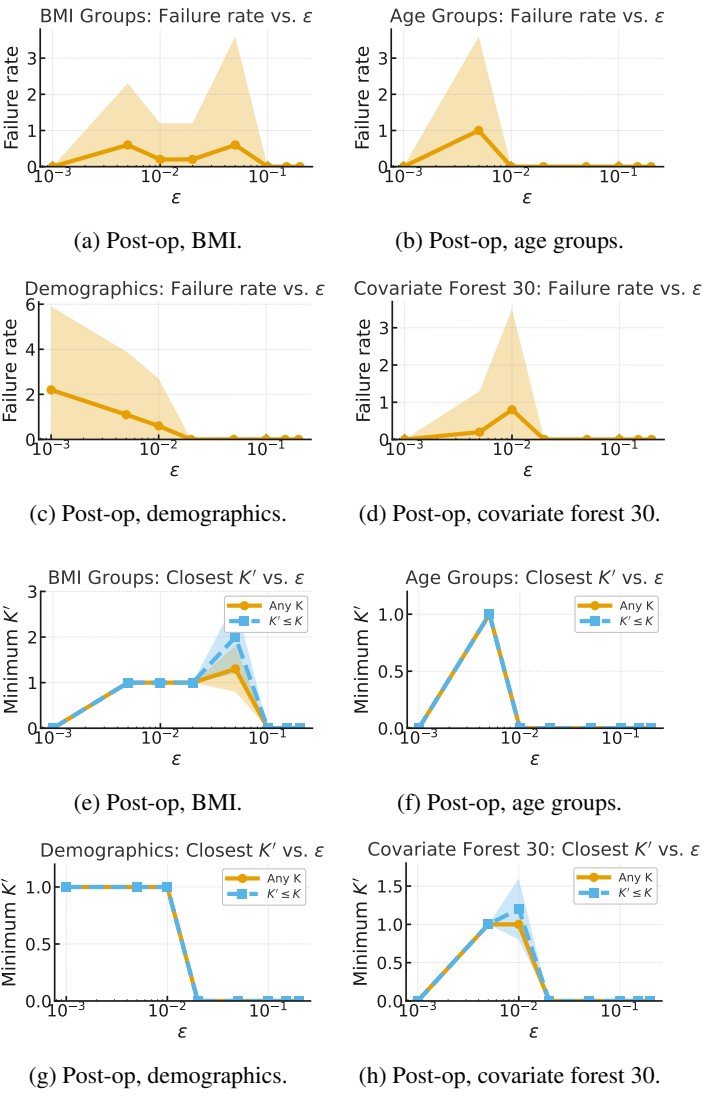

(a) Post-op, BMI.

(b) Post-op, age groups.

(c) Post-op, demographics.

(d) Post-op, covariate forest 30.

(e) Post-op, BMI.

(f) Post-op, age groups.

(g) Post-op, demographics.

(h) Post-op, covariate forest 30.

