# OpenReview forum: "Good Allocations from Bad Estimates"
_ICLR.cc/2026/Conference — ICLR 2026 Poster_

### Official Review · Reviewer_zW6P · 2025-10-29

**Soundness:** 3
**Presentation:** 3
**Contribution:** 3
**Rating:** 8
**Confidence:** 4

**Summary:**

The authors investigate sample complexity bounds for estimating CATE across M different groups when trying to perform allocation to a K subset of them. Standard CATE analysis demonstrates a $\frac{M}{\epsilon^2}$ bound, as each group needs to learn a separate CATE, then the K-highest CATE values are selected. However, the authors argue for a $\frac{M}{\epsilon}$ bound by performing coarser estimation for treatments near the boundary. Essentially, the authors demonstrate that if CATE estimates can be learned within $\sqrt{\epsilon}$, then any estimation mistakes will not be very costly. Such learning is possible under certain assumptions on $\tau$; for example, that $\tau$ is smooth or is near-uniform. The authors conclude with experiments on real-world RCTs to demonstrate the efficacy of their experiments.

**Strengths:**

1. **Work tackles an important problem** - The authors tackle the important problem of $K$ selection from $M$ groups in a causal setting. Such a problem can be seen across a variety of real-word situations, and is especially prevalent in the world of policy. This can help more efficiently allocate resources and avoid unnecessary experiments.
2. **Analysis is intuitive and clean** - The authors present a clean and intuitive reason why their proposed selector should outperform baselines. By avoiding the need to recompute CATE for each of the $M$ groups, the authors are able to achieve a better sample complexity, due to their ability to adaptively explore in some sense.
3. **Characterizes when their new bound is possible** - The authors describe several scenarios where there newly proposed estimators reach the desired $\frac{1}{\epsilon}$ bound, and describe why such assumptions are not onerous. For example, the authors describe a class of smooth distributions for $\tau$ that allow for such bounds, and they express an if and only if condition based on the CDF of $\tau$.
4. **Extension to Flexible Budget** - In Section 5, the authors include a discussion of flexible budgets, where an alternative budget $K'$ is used near $K$ that achieves better sample complexity performance. The authors sketch when such a method does and does not work, dependent on the distribution of $\tau$.

**Weaknesses:**

1. **Experiments are not Extensive** - The experiments in Section 6 are condensed to a half a page (with some extra material in the Appendix). As a result, it's hard to understand some of the results. For example, why is the failure percentage not monotonic in $\epsilon$; presumably with increasing $\epsilon$, it is less a stringent failure threshold, so it is surprising that this pattern is exhibited across datasets. Additionally, there is little comparison with the $\frac{1}{\epsilon^2}$ method. Finally, what does the actual distribution of $\tau$ look like in practice; are the assumptions validated?

**Questions:**

1. In practice, on the datasets listed in the experiments section, how does the sample complexity of the CATE-style selector ($\frac{1}{\epsilon^2}$ compare with the selector proposed

---

> ### Author Response · Authors · 2025-11-21
> **Rebuttal response (1/2)**
>
> We thank the reviewer for their positive and insightful comments.
>
> **Regarding the extensiveness of the experiments,** we agree that the experiments appear too condensed in the main body due to the lack of space. We provide all of the experimental details in Section E in the appendix. In particular, Section E.3 explains how we generate Figure 1 (and the similar figures in Section F generated with different groupings of each of the 5 RCT datasets) and Section E.4 how we generate Figure 2. We will try to explain them better in the main body.
>
> For Figure 1, we do not select a value of $\epsilon$; for each sample size and each budget, we sample from the population for that many samples and compute the treatment effect estimates for each of the $M$ units. Selecting the top $K$ coarse estimates yields an allocation, and we plot its realized value. Answering the reviewer’s question on the comparison to the $1/\epsilon^2$ method, the red line in these plots represents precisely the number of samples that we would expect from the worst-case bound of $O(M/\epsilon^2)$ (as per Lemma 3), whereas the green line represents the number of samples derived from our theoretical bound of $O(M/\epsilon)$, as per our Theorem 7. These plots demonstrate that, in practice, few samples suffice for an $(1-\epsilon)$-optimal allocation – much less than the $(1/\epsilon^2)$ sampler dictates.
>
> For Figure 2, we do select different values of $\epsilon$. We agree with the reviewer that expecting the failure percentage to be monotonic with $\epsilon$ is definitely the correct intuition. If one looks at Figure 2 along with all of the replicates for the different grouping methods in Section F (i.e., the two last figures in each of the subsections in Section F), one can see that the failure percentage is roughly monotonic with $\epsilon$, as one would expect, just with some small variations (the y-axis is re-sized to the max failure rate, so the few variations are all very small in magnitude). The main reason why we believe there are small variations is because the top $K$ decision boundary is discrete, and so which units “cross the boundary” based on their $\hat{\tau}(u)$ estimates with different $\epsilon$ values can fluctuate a bit and be swapped in a way that is not strictly monotonic with $\epsilon$ (it’s a discrete decision boundary, so there can be some variability). However, the overall trend is consistent. We also point out that there is a failure probability of delta, which can add some variability, and that in the experiments for this figure we account for *all* possible budgets from $K=1$ to $K=M$ in the computation of the failure rate, including the very small budgets (this can also cause some variability, especially in the cases where the total number $M$ of units is not very large). We will include these observations in the paper, and we can also include the plots where we do not account for the first few very small budgets.

---

> ### Author Response · Authors · 2025-11-21
> **Rebuttal response (2/2)**
>
> **Regarding the lack of comparison to the $1/\epsilon^2$ method,** we want to point out that we do consider it in the set of experiments relating to Figure 1 (and its various repetitions with other grouping methods) – the red line in Figure 1 represents the number of samples used by the $M/\epsilon^2$ selector. We agree that we should also compare it in the case of the set of experiments relating to Figure 2, and we will include it in the paper.  We can also give concrete numbers of the difference in number of samples between our method and the $M/\epsilon^2$ method for the values of epsilon that we test. If the reviewer is referring to any other types of comparisons to the $1/\epsilon^2$ method, please let us know.
>
> **Regarding what the actual $\tau$ distribution looks like in practice,** it can be instructive to include visualizations of the treatment effect curves in the RCT experiments for various grouping strategies – we are happy to include these in the paper. They generally look either Gaussian-like or uniform-like. As Figure 1 shows, the assumptions are indeed validated: we see that we obtain a close-to-optimal allocation (blue line) much quicker than the $M/\epsilon^2$ bound suggests (red line). In fact, even faster than our theoretically-proven bound for smooth distributions with $M/\epsilon$ many samples (green line).
>
> **Lastly, regarding the question on the sample complexity in practice,** as we explain above in this response, Figure 1 is meant to compare the $1/\epsilon$ and the $1/\epsilon^2$ methods in practice. Our proposed selector has sample complexity $O(M/\epsilon)$, as opposed to $O(M/\epsilon^2)$.
>
> In practice, a planner can benefit from our theoretical results in at least two ways:
> In one case, the planner has already decided on a fixed budget to use for the RCT. In this case, our results say that, whatever this budget is, it typically supports much better allocations than CATE estimates. Here the budget is fixed and the resulting error epsilon varies.
> In the second case, the planner has an epsilon-value in mind that they're hoping to achieve and is trying to decide on a budget (i.e., sample size) to use. Here, as per our main theorem, the planner can use $M/\epsilon$ many samples. If one wants to verify the optimality of the allocation for the chosen value of $\epsilon$, the planner can also use our optimality-certificate (which we show in Section 4.2). A different strategy for the planner, as explored in our Section 5 (and further in Section D), is to find an alternative budget $K’$, close to the original budget $K$, such that the optimality-certificate shows that the allocation is $\epsilon$-optimal, or where $\rho$-regularity around the budget threshold can be certified from the coarse estimates. As our experiments with real-world RCT data show, in all cases (for all 5 RCT datasets and all grouping strategies) we find that $M/\epsilon$ many samples suffice for a $(1-\epsilon)$-optimal allocation. In fact, we converge even quicker to an optimal allocation, as demonstrated in Figure 1. Hence, the experiments provide strong evidence that in typical instances (in other datasets) the second case succeeds in just one round.
>
> We again thank the reviewer for the sharp questions and we will make sure to include these various clarifications in the paper.

---

> > ### Comment · Reviewer_zW6P · 2025-11-22
> > **Reviewer Response**
> >
> > Thank you for your detailed rebuttal. I'm generally satisfied with this paper, and think it provides an important contribution. As such, I'll maintain my score of 8.

---

### Official Review · Reviewer_mQ5Z · 2025-10-31

**Soundness:** 2
**Presentation:** 2
**Contribution:** 2
**Rating:** 4
**Confidence:** 3

**Summary:**

The authors provide new theory showing that optimal treatment allocation can be achieved with fewer samples than classic predict-then-optimize approaches (FullCATE in the paper) would require. Their theory is built upon the insight that accurate estimates of CATE are only needed around the treatment allocation threshold.

**Strengths:**

- The authors provide interesting insights about sample size requirements for optimal treatment allocation.
- The authors substantiate their claims with extensive theoretical analysis

**Weaknesses:**

Some other related work exists that uses similar insights about the problem of optimally allocating treatment, though often without an extensive theoretical analysis. For example, some work has argued that when trying to find the optimal treatment allocation, accurate CATE estimation is not always the most effective [1,2].

While the authors provide a very extensive theoretical analysis, they only briefly explain the potential impact of their contributions. For example, I find it hard to understand what practitioners should do with this new information. It would be helpful if the authors could discuss this.

I wonder why the authors decided to put the individuals into different groups and do the analysis based on these groups. As the number of groups decreases, the number of samples needed also decreases, but the quality of the overall allocation will also go down (because as you make the groups more granular, you will find more heterogeneity between groups, allowing for better decision-making). How should you choose the number of groups in practice if you have very little a priori information about the CATE distribution?

Related to the previous point, I do not understand how the different groups are created in the experiments. To me, this seems like the most crucial part when evaluating treatment allocation quality.

I wonder why the authors did not use datasets that are often used in Uplift Modeling [1] and treatment effect estimation [3].

[1] Devriendt, F., Van Belle, J., Guns, T., & Verbeke, W. (2020). Learning to rank for uplift modeling. IEEE Transactions on Knowledge and Data Engineering, 34(10), 4888-4904.

[2] Fernández-Loría, C., & Provost, F. (2022). Causal decision making and causal effect estimation are not the same… and why it matters. INFORMS Journal on Data Science, 1(1), 4-16.

[3] Curth, A., Svensson, D., Weatherall, J., & Van Der Schaar, M. (2021, August). Really doing great at estimating cate? a critical look at ml benchmarking practices in treatment effect estimation. In Thirty-fifth conference on neural information processing systems datasets and benchmarks track (round 2).

**Questions:**

All the items listed in the Weaknesses section may be interpreted as questions by the authors.

Typo: line 353 "notion that requiring"

---

> ### Author Response · Authors · 2025-11-21
> **Rebuttal response (1/3)**
>
> We thank the reviewer for the insightful comments and feedback, which will definitely improve our paper, and for spotting the typo.
>
> We will include the related works brought up by the reviewer, which we agree are very relevant. As the reviewer points out, however, these two papers do not include any proven theorems, necessary and sufficient distributional conditions, or any lower/upper bounds, unlike in our work. We provide the first formal sample complexity separation between the problem of treatment allocation and the problem of CATE estimation. These papers provide more empirical observations that further validate our theoretical results. We will include a more detailed comparison in the related section of our paper. Thank you for pointing them out to us!
>
> We answer the questions raised in the Weaknesses section in order:
> 1. **The key take-away for practitioners is the following:** our work studies efficient algorithms for the goal of assigning a scarce resource among groups of a population (e.g., which schools in a city or groups of students should receive extra teachers). A typical way of doing so is through estimating the treatment effects of the groups to high accuracy (for example, through an RCT), and then assigning the resource among a subset of the groups – namely, the ones with the highest treatment effects (i.e., the ones who appear to benefit most from the treatment). Our results show that we don’t need to estimate these treatment effects up to accuracy epsilon in order to get a $(1-\epsilon)$-allocation; rather, $\sqrt{\epsilon}$ accuracy suffices. This is very relevant in practice because this means that we can obtain an $(1-\epsilon)$-accurate allocation of the scarce resource on the population with much fewer samples. Our theorems and work focus on studying the sample complexity of these algorithms: we want to obtain allocations of high accuracy, but using the fewest samples possible, because requiring more samples means requiring more resources and organization in practice, which can be very expensive.
>
> For example, in an RCT, the sample complexity savings translate into savings of the number of participants in the RCT. As we say in page 3, “There’s an immediate practical takeaway relevant to future policy decisions about targeting welfare-promoting interventions. The standard sample size calculations for CATE estimation are excessively pessimistic for the purpose of treatment allocation. Indeed, we can find nearly optimal allocations from coarse treatment effect estimates. Perhaps counterintuitively, an RCT that is severely underpowered for CATE estimation can still yield excellent allocations. As a rule of thumb, $M/\epsilon$ samples suffice for a $(1 − \epsilon)$-optimal allocation.”
>
> For example, if a government wants to run an RCT in order to determine which schools should get an extra teacher for the next academic year, then for $M$ total schools and a chosen value of $\epsilon$ (which represents the desired accuracy for the final allocation chosen by the government), the standard sample size calculations suggest that there should be $M/\epsilon^2$ many participants, where $\epsilon$ denotes the desired level of accuracy for the final allocation. Our theorems show that $M/\epsilon$ many participants suffice in all typical instances, and our experiments confirm that we converge to an optimal allocation with very few number of samples (Figure 1).
>
> We will emphasize this point more in our write-up, to ensure that our results can be more directly picked up by practitioners. For example, we can add a “For practitioners” subsection where we summarize our findings in a more practical way. Thank you for pointing out that this wasn’t clear enough.

---

> ### Author Response · Authors · 2025-11-21
> **Rebuttal response (2/3)**
>
> 2. **The reason we have groups** is because our work is placed within the setting of CATE estimation, which refers to conditional average treatment effect estimation. For a treatment $T$ (e.g., having an extra teacher or not), the CATE value for a unit $x$ is defined as $\tau(x) = \mathbb{E}[Y(1)-Y(0) | X=x]$, given covariates $X$. The reason we need groups is because the individual treatment effect cannot be computed directly, given that any one individual participating in an RCT gets assigned to either $T=1$ or $T=0$. Hence, we compute the average difference between the outcome of the individuals that have been assigned $T=1$ within unit $x$ and the outcome of the individuals that have been assigned $T=0$ within unit $x$. Hence, CATE is identifiable and can be estimated from observed data. This is why we have units/groups in our setting.
>
> 	Hence, the number of groups is decided before-hand by the social planner based on what it is that they are trying to allocate to the population, and so $M$ is fixed. In our work, we are interested in the optimal sample complexity given a set of groups. Our algorithms and results remain agnostic as to how the planner chooses the groups a *priori*. In a lot of cases, groups are defined by organizational units, like schools, hospitals, etc. In other cases, they may be chosen with domain knowledge, for example based on the features of the data.
>
> An interesting question, which the reviewer’s point seems to also indicate, is to study how varying $M$ changes the smoothness of the distribution of $\tau(x)$ values (because higher smoothness is preferable). We think that this would be an exciting future direction to explore.
>
> 3. **We detail the grouping methods** used to create the groups/units in all of the experiments in Section E.2 in the appendix – we apologize that we weren’t able to include it in the main body due to lack of space. In most cases, we create groups based on the covariates, adapted to what makes sense to each dataset and for what each RCT is measuring. For example, for the educational STAR RCT we group the students based on their baseline test performances. A different grouping with the STAR dataset that we do is to cluster the students based on school ID. In the case of NSW dataset, for example, which is in the domain of labor economics, we group the individuals based on the baseline earnings, on age groups, and so on (among other ways, we test multiple possible partitionings for each dataset). We also explore data-driven grouping methods, such as grouping individuals by their predicted CATE values.
>
> It is perhaps not exactly accurate to say that the grouping strategy is the most crucial part in evaluating the treatment allocation quality: our algorithm works for a fixed set of groups that is given as input to the algorithm, and what we are concerned with is whether $\rho$-regularity is satisfied in practice (because per our theoretical results we know that if $\rho$-regularity is satisfied, then we provably have an $\epsilon$-optimal allocation). So this is what we are testing with the various groupings, and we find that yes, our smoothness condition is met in practice for any a *priori* grouping, given that we do obtain optimal allocations with only $O(M/\epsilon)$ many samples (in fact, with even less samples, as Figure 1 shows).
>
> So, in other words, each grouping method gives us a distribution of $\tau(x)$ values, given that the $\tau(x)$ is a value on a group $x$ (because we are working in the setting of CATE estimation). We find that, in practice, each such distribution is smooth and thus gives us good allocations with coarse estimates. The number of groups is fixed for each partitioning, but can be varied (e.g., when we do clustering we try different numbers of groups). $M$ is fixed for each of the plots that we present.

---

> ### Author Response · Authors · 2025-11-21
> **Rebuttal response (3/3)**
>
> 4. **We thank the reviewer for bringing up these datasets to us.** We want to emphasize that the datasets that we use in this paper (the STAR dataset, the TUP dataset, the NSW dataset, the acupuncture dataset, and the postoperative dataset) have extensively been used in the causal inference literature and treatment effect estimation, as we cite in our paper. These datasets appear in many causal inference papers (each of these appear extensively used by researchers in Google Scholar, all along from the year the study was run up until 2025). We are happy to include the datasets mentioned by the reviewer in our final version – we agree with the reviewer that these other datasets in the provided references are also widely used in research studies.
>
> We do want to point out that the goal of our experiments is to confirm our theoretical results, which say that we get good allocations with very few samples. In all 5 cases, along with all of the different grouping methods, we confirm that this is indeed the case (see Figure 1 and all of the other runs with different grouping methods presented in Section F). Given the uniformity of the pattern, using these five datasets appeared to be enough to us (and especially given that we chose the RCTs from very different domains, from healthcare to education). It also made more sense to focus on RCT datasets where enrolling participants is more costly and hence where one would want to be able to work on small data regimes, given that our goal is to reduce the sample complexity needed to get a good allocation and hence reduce the cost of the experiments.
>
> We thank the reviewer again for their detailed feedback and questions!

---

> > ### Comment · Reviewer_mQ5Z · 2025-11-21
> >
> > Thank you very much for your clarifications! This has really improved my understanding of your contribution. I believe such a "for practitioners" paragraph/section would increase the quality of the paper.
> >
> > Given my improved understanding, I will also raise my score.
> >
> > Now that I better understand how the grouping works, I have one final question (related to why I initially thought the grouping choice so important) to confirm my understanding. If you were to choose the groups at random (so not based on covariates, but of course, there will be small differences between the groups due to sampling), would that mean that $\rho$-regularity is no longer satisfied? What would happen to the empirical results? This type of grouping would probably not make sense in practice, but I think this would provide some additional intuition.

---

> > > ### Author Response · Authors · 2025-11-28
> > >
> > > Thank you very much for reading our response and for the follow-up questions and comments. We are glad that we were able to aid in the understanding of our work. We completely agree and will add a "for practitioners" section in the final version of the paper.
> > >
> > > Yes, that is the correct intuition -- if the groups are chosen at random, then the CATE values are expected to cluster around the average treatment effect value of the population, with variance inversely proportional to the number of groups $M$. In this case, we wouldn't expect $\rho$-regularity to hold, as the reviewer correctly indicates, but our low-accuracy allocation should still obtain a near-optimal allocation, for the reasons described in Section 5 of our paper. Namely, if the CATE values are essentially all equal to one another, then the value of the allocation remains close to optimal, even if we select many incorrect units (precisely because their CATE values are very close to each other). However, as the reviewer indicates, such a grouping wouldn't really make sense in practice, but we agree that this intuition is important and will add a lengthier explanation in the paper about how the groups are defined and how the chosen groups impact the distribution of CATE values and the subsequent optimality of the allocation. Moreover, the cases where $\rho$-regularity isn't satisfied (due to a high clustering of the units around one value) possess a much higher degree of arbitrariness, regardless of which allocation algorithm is used on the units. This is another reason why $\rho$-regularity is a desirable property, and why we encourage practitioners to find a slightly different cut-off point $\tau_K$ in such cases (i.e., use a "flexible budget"), so that the cut-off point lies in an area with low mass.
> > >
> > > Thank you again for your comments and engagement!

---

### Official Review · Reviewer_92Sv · 2025-10-31

**Soundness:** 4
**Presentation:** 4
**Contribution:** 4
**Rating:** 6
**Confidence:** 4

**Summary:**

This paper fills the gap between estimating CATE and making decisions on the allocations. The authors show that while estimating all CATEs within $\epsilon$ accuracy requires $O(M/\epsilon^2)$ samples, achieving a near-optimal $(1-\epsilon)$ treatment allocation typically needs only $O(M/\epsilon)$ samples under mild distributional assumptions. In general, I personally see the results quite interesting and insightful.

**Strengths:**

1. The paper makes a clear and elegant theoretical distinction between estimation and allocation. The reduction of the sample complexity from $M/\epsilon^2$ to $M/\epsilon$ is insightful and exciting.
2. Practical relevance: Direct implications for RCT and policy design: significant reduction in sample cost.
3. The proofs are clean and well-structured and the theoretical results are rigor.

**Weaknesses:**

In general, I enjoy reading the paper a lot. I do not have major concerns.

1. Comparison to bandit best arm identification could be expanded. The link is conceptually strong. Particularly, recently, there are some works on good arm identification. Some ideas are very similar, although they are not is a causal inference setting.

2. Policy implication is strong (“RCTs underpowered for CATE estimation can still yield good allocations”), but guidance on how to detect $\rho$-regularity or compute sample sizes in practice is missing.

**Questions:**

See above.

---

> ### Author Response · Authors · 2025-11-21
> **Rebuttal response (1/2)**
>
> We thank the reviewer for their positive feedback, particularly in highlighting that they find our main results insightful and exciting!
>
> **Regarding the comment on the comparison to bandit best arm identification,** we agree completely with this point and we will add an expanded related work section in the appendix to highlight the algorithms in the literature relating to good arm identification. In particular, we will make sure to include more works on variants of Top-K arm identification, good-arm identification, and thresholding bandits from the multi-armed bandit literature. Besides the ones that we already cite in our paper, some others include: “Optimal PAC Multiple Arm Identification with Applications to Crowdsourcing” by Zhou et al. 2014, “Learning with Limited Rounds of Adaptivity: Coin Tossing, Multi-Armed Bandits, and Ranking from Pairwise Comparisons” by Agarwal et al. 2017, “On Top-k Selection in Multi-Armed Bandits and Hidden Bipartite Graphs” by Cao et al. 2015, “Nearly Instance Optimal Sample Complexity Bounds for Top-k Arm Selection” by Chen et al. 2017, and “A Bandit Approach to Multiple Testing with False Discovery Control” by Jamieson et al. 2018, among others. We will discuss these papers as well.
>
> As the review points out, while some of the intuition behind these top-K papers are similar to the intuition as to why we can achieve an optimal allocation with coarser estimates, our causal inference set-up focused on CATE estimation is different and key to the formalization of our set-up, where we consider non-adaptive sampling consistent with RCT design (rather than sequential bandits) and focus on the distribution of treatment effect values and its smoothness properties (unlike in the bandits literature). We will also discuss more how the usual $\Delta$ term in the bandit literature (which measures the smallest separation between the means of the arms) relates to our $\rho$-regularity condition, and how our flexible budget results could be applied to by-pass some of the known lower-bounds that use the $\Delta$ term in the bandit literature, where underspending/overspending on the parameter $K$ is usually not considered. We hope that our analysis of the flexible budget approach in Section 5 could in turn be beneficial to the bandit literature in by-passing the usual $1/\Delta^2$ lower bound in the sample complexities.

---

> ### Author Response · Authors · 2025-11-21
> **Rebuttal response (2/2)**
>
> **Regarding the second point on practical guidance,** we want to emphasize that in Section 4.2 (and continued in Section C.2 due to lack of space) we provide provable certificates to check for $\rho$-regularity and for the optimality of the obtained allocation. The theoretical results in this section precisely provide guidance on how to check whether $\rho$-regularity is achieved in practice from the coarse estimates (Claim 3.4 & Section C.2). Alternatively, Claim 4.1 (continued in Section C.2) provides a provable certificate, using the coarse estimates, that checks whether we have achieved an $(1-\epsilon)$-optimal allocation. We will make these two points clearer in the paper; we completely agree that these are important.
>
> Regarding the guidance on how to compute sample sizes in practice, we imagine that a planner might benefit from our results in at least two ways:
> 1. In one case, the planner has already decided on a fixed budget to use for the RCT. In this case, our results say that, whatever this budget is, it typically supports much better allocations than CATE estimates. Here the budget is fixed and the resulting error epsilon varies.
> 2. In the second case, the planner has an epsilon-value in mind that they're hoping to achieve and is trying to decide on a budget (i.e., sample size) to use. This is where our optimality-certificate comes in handy. The planner can start by taking, say, $2M/\epsilon$ samples. If the optimality-certificate shows that the resulting allocation is $(1-\epsilon)$-optimal, the planner is done. Otherwise, the planner may double the sample size and repeat. Once the optimal sample size is reached (up to a factor 2), the optimality certificate will succeed.
>
> A different strategy for the planner, as explored in our Section 5 (and further in Section D), is to find an alternative budget $K’$, close to the original budget $K$, such that the optimality-certificate shows that the allocation is $\epsilon$-optimal, or where $\rho$-regularity around the budget threshold can be certified from the coarse estimates.
>
> We will make these points clearer in the paper -- we can add a section on "Practical recommendations". As our experiments with real-world RCT data show, in all cases (for all 5 RCT datasets and all grouping strategies) we find that $M/\epsilon$ many samples suffice for an $(1-\epsilon)$-optimal allocation. In fact, we converge even quicker to an optimal allocation, as demonstrated in Figure 1. Hence, the experiments provide strong evidence that in typical instances we expect the second case to succeed in just one round, and so the planner can go ahead with $M/\epsilon$ many samples (where $\epsilon$ is the desired accuracy of the final allocation).
>
> We envision that future work could make this certificate more granular by, for example, adding some more adaptive rounds to our algorithm, by combining our results with tools from the property testing literature, or by incorporating domain knowledge that the social planner has a *priori* about the population.
>
> We thank the reviewer again for their comments and questions.

---

> > ### Comment · Reviewer_92Sv · 2025-11-21
> > **Thank you for the clarification**
> >
> > Thank you for your clarification and response. I will be very happy to increase my score to 7 if there is such a choice (but there isn't). Good luck!

---

### Author Response · Authors · 2025-12-03
**Summary of the rebuttal period**

We thank the reviewers for their insightful comments and engagement. We write this summary of the rebuttal period to the new AC assigned to this submission, as per ICLR's response to OpenReview's unfortunate leak.

- Reviewer 92Sv finds that our paper "makes a clear and elegant theoretical distinction between estimation and allocation", and that the sample complexity separation that we show is "insightful and exciting". Similarly, Reviewer zW6P expresses that it "provides an important contribution", and Reviewer mQ5Z that the results provide "interesting insights". All reviewers emphasize that the problem we study is important and relevant in practice, due to the significant reduction in sample cost achieved by our algorithm, and agree that the theoretical results are rigorous. Reviewer 92Sv found them "clean and well-structured", Reviewer mQ5Z emphasized that the "authors substantiate their claims with extensive theoretical analysis", and Reviewer zW6P said that the "analysis is intuitive and clean".

- All reviewers pointed out that we should explain the policy implications better, and we suggested adding a section on "guidance for practitioners" as a response, which the reviewers agreed would be valuable. We clarified the experimental set-up to Reviewer mQ5Z and Reviewer zW6P, regarding how the groups are formed experimentally to the former and how exactly we do a comparison to the $1/\epsilon^2$ method in our figures to the latter. We agreed to explain these two key points better, and to expand on the description of the experimental set-up in the main body.

- Reviewer 92Sv and zW6P further indicated that they do not have any major concerns. After our rebuttal responses, Reviewer 92Sv said that they would be "very happy" to increase their score to 7 if it was an option, and Reviewer mQ5Z said that our rebuttal clarifications helped in their understanding ("really improved my understanding of your contribution") and positive evaluation of the paper ("Given my improved understanding, I will also raise my score."). Reviewer zW6P maintained the score to an 8, saying that they are "generally satisfied with this paper".

We thank everyone again for engaging with our work.

---

### Meta-Review · Area_Chair_wxa2 · 2026-01-06

**Summary:**

This paper develops methods to improve the sample complexity required for estimating the conditional average treatment effect (CATE).

Reviewer 82Sv evaluated the paper very positively and raised questions regarding possible extensions to best-arm identification, as well as practical guidance and sample size calculations. In response, the authors added additional survey results and explained the similarities and differences between their setting and best-arm identification. They also provided formal proofs and clarified which practical procedures are feasible in real applications. The reviewer indicated that these responses were satisfactory (within the rebuttal period) and expressed agreement with the authors’ explanations.

Reviewer mQ5Z asked about the practical use of the proposed contributions and the validity of partitioning individuals into different groups. Relatedly, the reviewer raised questions about the applicability of commonly used datasets. The authors responded by proposing concrete practical use cases and by adding a new section that provides explicit guidelines. They also offered a detailed justification for the grouping strategy. The reviewer stated that they were satisfied with these clarifications and indicated an intention to raise their score.

Reviewer zW6P gave a strong overall evaluation of the paper but noted that the experimental evaluation was not fully comprehensive. Taking this feedback into account, the authors explained additional experimental results that could not be included in the main text and provided further clarification of the experimental settings. The reviewer indicated that these explanations resolved their concerns and likewise raised their score.

Overall, because the authors responded promptly and thoroughly during the rebuttal phase (rather than the discussion phase), all reviewers indicated that their concerns had been resolved and that they would increase their scores. Combined with the fact that the paper’s contributions were already strong, and that the rebuttal allowed for sufficient clarification and discussion, the paper can be considered to have achieved a clear consensus in favor of acceptance.

**Reviewer Concerns:**

See above.

**Reviewer Scores:**

See above.

---

### Decision · Program_Chairs · 2026-01-26

Accept (Poster)